# Brief communication: A framework to classify glaciers for water resource evaluation and management in the Southern Andes

Nicole Schaffer[1], Shelley MacDonell[1, 2]

[1] Centro de Estudios Avanzados en Zonas Áridas (CEAZA), ULS—Campus Andrés Bello, Raúl Bitrán 1305, La Serena, Chile
[2] Waterways Centre for Freshwater Research, Lincoln University and the University of Canterbury, Private Bag 4800, Christchurch, New Zealand

*Correspondence to*: Nicole Schaffer (nicole.schaffer@ceaza.cl)

**Abstract** Over the last two decades the importance of Andean glaciers, particularly as water resources, has been recognized in both scientific literature and in the public sphere. This has led to the inclusion of glaciers in the Environmental Impact Assessment and the development of Glacier Protection Laws in both Chile and Argentina. However, these laws are limited in their ability to protect, manage, and monitor water resources as they do not differentiate between glacier types. We propose three glacier categories that aim to group glaciers based on their sensitivity to environmental changes as a framework that could be adopted to match the level of protection to the current and future needs of society, be region-specific, and could evolve through time. Finally, we review both national inventories with respect to this classification to facilitate the evaluation and/or management of water resources.

## 1 Introduction

Over the last two decades, the role of glaciers in the headwaters of Andean basins has become increasingly prominent in both scientific literature and in the public sphere from the community level to national public policy. This interest has been motivated primarily by the increased awareness of climate change impacts and other environmental considerations (Herrera Perez and Segovia, 2019; Jones et al., 2018; Masiokas et al., 2020). This has led to the development of Environmental Impact Assessment (EIA) measures specifically designed for glaciated regions, and the development of glacier protection laws (GPL) that aim to preserve glaciers as strategic water reserves, for their role in sustaining biodiversity, in sustainable tourism, and their scientific importance (Gobierno de Argentina, 2010; Senado de Chile, 2019). Both Chile and Argentina have funded the creation of detailed national inventories (Barcaza et al., 2017; Zalazar et al., 2020) and detailed glacier monitoring plans (CECS, 2009; IANIGLA-CONICET, 2019). At a local scale, there is an acknowledgment by councils and municipalities as well as community groups, that there is a need to better understand the behaviour and characteristics of glaciers to better manage water supplies. For example, in Chile this has led to regional governments funding studies of glacier distribution (García et al., 2017) and management plans (MacDonell and González, 2019). Andean glaciers are also landmarks of national heritage and have important cultural and indigenous significance (Bosson et al., 2019; National Geographic, 2021). Despite the recognized importance of Andean glaciers, current (or proposed) EIA protocols and GPL in Chile and Argentina are limited

in their ability to protect, manage, and monitor these water resources as they do not differentiate between glacier types. Currently, many of the requirements in the EIA process ([www.sea.gob.cl)](www.sea.gob.cl) are the same regardless of glacier type, and variable impacts are not given adequate consideration. For example, a debris-free glacier would be more sensitive to air particles such as black carbon from a nearby road than a debris-covered glacier, but this difference cannot be adequately addressed within the current EIA. For the EIA as well as when generally considering the protection, evaluation, and management of glaciers as water resources it is important to consider that different glacier types may have distinct sensitivities. Here we define sensitivity as change in mass balance over a given period of time in response to environmental changes (e.g. changes in temperature or precipitation).

Traditionally glaciers have been grouped into three categories: debris-free glaciers, debris-covered glaciers and rock glaciers. However, these categories do not necessarily reflect the sensitivity to environmental factors (e.g. climate) and are often difficult to implement for practical applications. The distinction between debris-free and debris-covered glaciers is relatively well defined in the literature, however in practice, often a more precise dividing line is needed. Furthermore, the division between a debris-covered glacier and a rock glacier is often ambiguous. In some instances glaciers that have a very thin debris cover and some ice exposed are considered rock glaciers (e.g. Chilean national inventory), while in other cases a thick enough debris cover to insulate the ice below is required ($\sim$> 3 m; Janke et al., 2015). The difference between these interpretations is an important consideration since the former option potentially encompasses glaciers that have a debris cover thin enough to allow sufficient heat transfer to melt the ice surface below (e.g. < $\sim$0.2 m; Nicholson and Benn, 2006), while the latter option only includes glaciers that have a thick enough debris cover to insulate them from changes in temperature at the surface (Bonnaventure and Lamoureux, 2013; Janke et al., 2015). In theory these glaciers with a very thick debris cover are less sensitive and therefore act as longer-term water reservoirs (Jones et al., 2018). To ensure an appropriate level of protection, monitoring program or management strategy is applied, it is useful to evaluate where these dividing lines should be and why as a first step towards creating classifications that reflect glacier sensitivity. This is particularly important when evaluating water resources over decadal or longer timescales. The classifications also provide a basis for discussion and will likely be of practical use for legislation and management.

The overarching goal of this paper is to propose an ideal dividing line (debris thickness) between each glacier category, account for additional factors that may impact sensitivity (see Sect. 3), and combine these to classify glaciers in a way that reflects their sensitivity to environmental changes (e.g. temperature and precipitation). We undertake a thorough evaluation of the Chilean and Argentinian national inventories to determine if they align with the proposed groups. Based on this, suggestions are provided to modify these inventories to facilitate the evaluation and/or management of water resources associated with the cryosphere in the semiarid Andes in the first instance.

The appropriate dividing line will vary from north to south along the Andes given the large variation in climate, topography,
and glacier characteristics (CECS, 2009; Masiokas et al., 2020). This variability is recognized within the national glacier
strategy for Chile (CECS, 2009) which identifies four distinct zones for glacier monitoring within which these three factors
are relatively homogeneous. The most northern zone (Zona Norte: 18-32°S) has numerous peaks above 5000 m and is arid,
resulting in relatively small glaciers at high altitude. Southward the precipitation increases, and the snowline drops in elevation.
The central zone (Zona central: 32-36°S) is also characterized by high peaks, but the snowline is lower giving rise to larger
glaciers that extend from mountain summits to valley bottoms. In the southern zone (Zona Sur: 36-46°S) the elevation of the
Andes Mountains drops, and glaciers are reduced to isolated volcanic cones. In the most southern zone (Zona Austral: 46-
56°S) the elevation of the Andes Mountains increases while the snowline continues to drop giving rise to large glaciers and
icefields that extend to sea level. The water supply from mountains compared to the entire basin also varies from north to
south. This "water tower" supply index has been calculated globally and in northern Chile it is 0.15 while in Southern Chile
where glaciers are much larger it is 0.34 (Immerzeel et al., 2020). We have chosen to focus the classification on the semiarid
Andes (~27°-35° S) which encompasses the transition between the most northern and central zones. This area is particularly
relevant for water resource evaluation, legislation and management given that it is water-scarce (DGA, 2016), many glaciers
are outside of protected areas (SNASPE for Chile, Areas Protegidas for Argentina), and it has a relatively high population
density. For example, in the semiarid Andes of Chile only ~10% of glacier surface area lies within protected areas, compared
to ~89% south of 35° where there is sufficient water availability (calculations completed using the 2014 Dirección General de
Aguas (DGA) glacier inventory accessible at https://dga.mop.gob.cl/estudiospublicaciones/mapoteca/Paginas/Mapoteca-
Digital.aspx). The classification proposed for the semiarid Andes is meant to serve as an example upon which classification
schemes for other regions could be based.

In the semiarid region the mean annual glacier contribution to streamflow varies from ~3-44 % for most years and can be
> 65 % during dry periods (Ayala et al., 2016; Schaffer et al., 2019). Rock glaciers are well insulated from the environment
by a thick debris cover and while their contribution per unit area to annual streamflow is likely to be less than other glacier
types, they may provide an important contribution at the end of summer (e.g. > 10 %; Schaffer et al., 2019; Schrott, 1996) and
also act as longer-term reservoirs (Jones et al., 2018; Schaffer et al., 2019).

## 2 Defining debris-covered glaciers and rock glaciers

In general, a glacier is defined as a perennial mass of ice (or perennially frozen ice and debris in the case of rock glaciers)
showing evidence of past or present flow detectable in the landscape by the presence of front and lateral margins (Cogley et
al., 2011; Delaloye and Echelard, 2020). A debris-covered glacier has a debris layer that varies in thickness with ice exposed
at the surface due to the discontinuity of debris cover or thermokarst depressions among other features (Janke et al., 2015;
Monnier and Kinnard, 2017). Thermokarst is a terrain-type characterized by irregular surfaces including hollows such as ice

collapse features. Some debris-covered glacier definitions require that most of the ablation zone be covered by debris (Barcaza et al., 2017; Cogley et al., 2011). Other definitions specify that the glacier may be fully covered (Delaloye and Echelard, 2020). Rock glaciers are defined as having a debris cover that is thicker than debris-covered glaciers and a discernible frontal slope that is generally convex (Delaloye and Echelard, 2020; Janke et al., 2015; Monnier and Kinnard, 2017). Some definitions

specify that the debris cover must be thick and continuous enough so that in general no ice is exposed at the surface (typically several meters thick; Janke et al., 2015; Monnier and Kinnard, 2017; Schaffer et al., 2019). Other definitions specify that debris must cover the entire glacier or differentiate debris-covered glaciers from rock glaciers by the presence of visible ice on the former, implying that no ice is visible on rock glaciers (Barcaza et al., 2017). These definitions for debris-covered and rock glaciers have been sourced from publications on the Andes, to ensure the definitions are locally relevant.

In summary, debris-covered glaciers are defined in the literature as being partially to fully covered by debris. Rock glaciers are defined as generally having no ice visible at the surface. While these definitions are suitable for scientific investigation, they are not sufficient for water resource management as they do not effectively differentiate between debris-covered glaciers that are sensitive to environmental changes (e.g. temperature, precipitation) compared to those that are not.

**3 Glacier classification for water resource management**

If the categories of glacier types are to differentiate between glaciers that have different sensitivities to changes in the environment (e.g. temperature and precipitation), then debris cover thickness must also be considered, since this has an important influence on glacier melt patterns (Ayala et al., 2016; Burger et al., 2019). Measurements from glaciers in the Himalaya, Canada, and Sweden have shown that a very thin debris cover (<~2 cm) results in higher melt rates than debris-free

glaciers due to a reduction in albedo and that under thicker debris cover melt rates progressively decline (Nicholson and Benn, 2006; Östrem, 1959). Heat continues to be transferred through the debris, resulting in surface melt, even when the debris cover is more than a couple of decimetres thick. For example, on Pirámide Glacier (33.57° S, 69.89° W) the debris thickness varies from 0.2 to 1 m and in areas where it is 0.2 to 0.3 m there is sufficient heat transmitted through the debris layer to result in ice melting at the surface throughout the day (Ferrando, 2012). Ayala et al., (2016) estimated the debris thickness and modelled

glacier mass balance on Pirámide Glacier. From the highest elevations, mass balance becomes more negative as elevation decreases as would be expected, until ~3800 m.a.s.l, below which debris cover thickens, and the mass balance suddenly becomes less negative and remains constant down-glacier (~-1 m w.e. a-1). The debris thickness at 3800 m a.s.l. is heterogeneous with a range of approximately 0.1-0.5 m thick (modelled debris thickness). Plots of modelled debris thickness versus mass balance show that on Pirámide ablation is reduced by 80% when debris thickness is 30 cm and 90% when it is 60

125  cm (A. Ayala, personal communication, March 7 2022). Estimated debris thicknesses > 0.2 m in this study under-estimate compared to in situ measurements and are prone to error so these results should be interpreted with caution. This agrees with Rounce et al. (2021) who provide globally distributed debris thicknesses and sub-debris melt outputs and conclude that thin

debris cover (typically 0.03 m – 0.05 m) enhances sub-debris melt while thick debris cover can result in a >90% reduction in sub-debris melt. We suggest that a thickness of > ~0.5 m could be used as a threshold between glacier classifications for the semiarid Andes since surface melt appears to be strongly reduced by debris cover above this threshold at Pirámide Glacier. According to Janke et al. (2015) a fully covered glacier (about 95% of the surface) often has a debris thickness of 0.5 – 3.0 m. Therefore, having > 95 % of the surface or more covered by debris could be used as a criterion to approximately identify this threshold using satellite imagery. Global products of glacier debris cover could be used to quantify the percentage of debris cover to remove subjectivity (e.g. Herreid and Pellicciotti, 2020; Scherler et al., 2018), however outputs have not been validated for the Andes and coverage is limited to glaciers included in the Randolph Glacier Inventory (RGI). We proposed that this initial classification could be refined or used in combination with modelled debris thicknesses (e.g. Rounce et al., 2021) but not replaced by these model outputs since validation in the Andes is needed and coverage is limited (see Sect. 5).

A thickness ~>3 m is required to thermally insulate the ice within the glacier and preserve the ice structure (Janke et al., 2015). For example, at Llano de las Liebres rock glacier (30.25° S, 69.95° W), seasonal variations in temperature affected ground temperatures between 2 to 5 m depth (Janke et al., 2015). When the debris cover is thick enough to preserve the ice structure, the surface is relatively smooth since the degradation of ice leading to the formation of thermokarst depressions is no longer actively occurring (Janke et al., 2015).

We suggest three categories for glacier classification for the purpose of water resource evaluation and/or management within the semiarid Andes (~27°-35° S; see Fig. 1 for examples):

1. Glaciers that are likely sensitive to environmental changes. These glaciers have exposed ice and include debris-free and some debris-covered glaciers (Fig. 1a).

2. Intermediate glaciers defined as having > 95 % debris coverage and a rough surface due to the discontinuity of debris cover, thermokarst depressions including "fresh" ice collapse features, or other features. We define "fresh" ice collapse features as depressions with at least one steep side that creates an abrupt change in topography, usually filled with water, ice or snow (Fig. 1a,b). We assume that the presence of "fresh" collapse features indicates that the glacier is somewhat sensitive to climate as such thermokarst features may be a sign of degradation at depth in the glacier (Schrott, 1996).

3. Glaciers that are likely thermally insulated from the environment (Fig. 1c). Based on examples in Janke et al. (2015) and our own observations of more than one hundred glaciers in the semiarid Andes of Chile and Argentina with high resolution satellite imagery (see Supplementary material: Inventory area reviewed.kmz), we conclude that these glaciers generally have no exposed ice, convex topography, a discernible frontal slope, and thermokarst depressions are uncommon and generally appear "weathered". "Weathered" depressions have sides that appear eroded and do not form an abrupt change in topography (Fig. 1b). These are definitively rock glaciers.

Insulated glaciers (Category 3) may have pronounced ridges and furrows perpendicular to the direction of flow, while intermediate glaciers (Category 2) have either no ridges or weakly developed ridges. Differentiation between intermediate and insulated glaciers could be improved by using both the qualitative classification proposed and modelled debris thicknesses, although these model outputs have large uncertainties (see Sect. 5). Insulated glaciers should not include rock glaciers that no longer contain ice (relict rock glaciers), however we recognise that such features may still play a significant role in the local catchment by enhancing liquid water storage and delaying spring runoff (Winkler et al., 2016). These may be differentiated from other glaciers by their collapsed appearance, often shallow or eroded frontal slope, and if necessary, confirmed using geophysical techniques. Some glaciers may present individual exceptions to the above guidelines and would need to be evaluated on a case-by-case basis.

The theory that glaciers with little to no debris cover should be more sensitive than those mostly covered by debris appears to hold true for the La Laguna catchment, where the Tapado Glacier is located. Robson et al. (2022) computed the elevation change for this catchment for 2012-2020 using combination of historical aerial photography, stereo satellite imagery, airborne lidar, and the Shuttle Radar Topography Mission (SRTM) DEM. The debris-free section of Tapado (Fig. 1a) shows the greatest elevation change by far with an average loss of -0.65 m a$^{-1}$, while the vast majority of debris-covered glaciers outside of the Tapado Glacier complex had either no detectable change or a surface lowering of $< 0.03$ m a$^{-1}$. Several of these debris-covered glaciers showed modest surface lowering rates as high as $> 0.1$ m a$^{-1}$. This agrees with a global study by Rounce et al. (2016) who conclude from their globally distributed debris thicknesses and sub-debris melt outputs that the net effect of accounting for debris in all regions is a reduction in sub-debris glacier melt, by 37% on average. Furthermore, Ayala et al. (2016) and Ferguson and Vieli (2020) expect debris-covered glaciers to react more slowly to a changing climate.

However, this does not hold true everywhere in the semiarid Andes nor in the world. For example, Ayala et al. (2016) report similar mass losses for Pirámide glacier (classified as intermediate) and two nearby debris-free glaciers, mainly because Pirámide is at a lower elevation. Similar mass loss rates for debris-covered and debris-free glaciers or parts of these in High Mountain Asia have also been observed (Gardelle et al. 2013; Kääb et al., 2012). The presence of supraglacial lakes, ice cliffs, reduced velocities at the tongue are thought to be responsible for a considerable increase in overall glacier mass loss (Pellicciotti et al., 2015; Ayala et al., 2016; Ferguson & Vieli, 2020; Rounce et al., 2021). These factors and/or thin debris cover are proposed to explain the similar mass loss rates. Given that debris-covered glaciers in this region and elsewhere can have similar mass balance rates as debris-free glaciers, we suggest a conservative approach when assigning a level of sensitivity for protection to intermediate glaciers by initially assuming they will have the same mass balance rate as sensitive glaciers, with the option to downgrade this if there is data available to justify the change.

In general when assigning a category for protection, we assume that a glacier made up of multiple glacier types (Fig. 1a) is hydrologically connected and therefore a disturbance of one part will impact the entire system and the water quantity and

quality downstream. We therefore suggest the same level of protection be applied to the entire glacier. In most cases where multiple glacier-types are present, the level of protection and monitoring associated with the most sensitive category should be applied (e.g. Fig. 1a and 2a, Table 1). However, where this part of the glacier is very minor (< ~20% of the surface area), it may be more appropriate to use the second most sensitive glacier classification instead (Fig. 2b,c). The initial category for

protection would be sensitive for glaciers that include either sensitive (class 1) or intermediate (class 2) glaciers and insulated for class 3 glaciers (Table 1). When more information becomes available, the sensitivity level can be downgraded if justified. High resolution datasets of glacier elevation change (e.g. Braun et al., 2019; Hugonnet et al., 2021; Robson et al., 2022) or modelled mass balance informed and/or validated with in situ data (e.g. Ayala et al., 2016) could be used to roughly determine the category for protection of an intermediate glacier. The specific mass balance (mass balance per unit area) could be

compared to that of sensitive and/or insulated glaciers nearby. If closest to a value between sensitive and insulated glaciers, the category for protection would be changed to intermediate. If closest to that of nearby insulated glaciers it would be changed to insulated. Examples are provided in Table 2.

Model outputs for sensitive and some intermediate glaciers in the Southern Andes (south of ~25° S) show the vast majority of

these glaciers have already reached or are expected to reach their maximum runoff or "peak water" before 2050 with a decrease in runoff thereafter (Burger et al., 2019; Huss and Hock, 2018). Insulated glaciers (rock glaciers) are more resilient to changes in temperature and therefore provide long-term water reservoirs (Bonnaventure and Lamoureux, 2013; Jones et al., 2018). However, this resilience can be diminished with human intervention such as the construction of roads or deposition of waste material on these glaciers, potentially leading to slope instability and permafrost degradation (Brenning and Azócar, 2010). As

well as contributing water, these glaciers likely play a role in storing and delaying runoff by several months (Winkler et al., 2016). Sensitivity may reflect runoff with more sensitive glaciers contributing more to streamflow compared to insulated glaciers, but there is not enough information to form conclusions for the semiarid Andes at this time (Schaffer et al., 2019).

Whilst debris cover impacts thermal properties, it may also mitigate the impact of precipitation changes. Ayala et al. (2016)

found that the mass balance sensitivity of the debris-covered glacier Pirámide was considerably lower than two adjacent debris-free glaciers. Thus, debris-covered glaciers may be less sensitive to both temperature and precipitation in this region. We suggest further investigation on this topic given that debris-free glaciers in northern Chile are known to be very sensitive to changes in precipitation (e.g. Kinnard et al., 2020), predominantly due to associated changes in surface albedo (e.g. MacDonell et al., 2013). Whilst not explicitly stated in the above definitions, the proposed classifications should therefore account for

precipitation sensitivity.

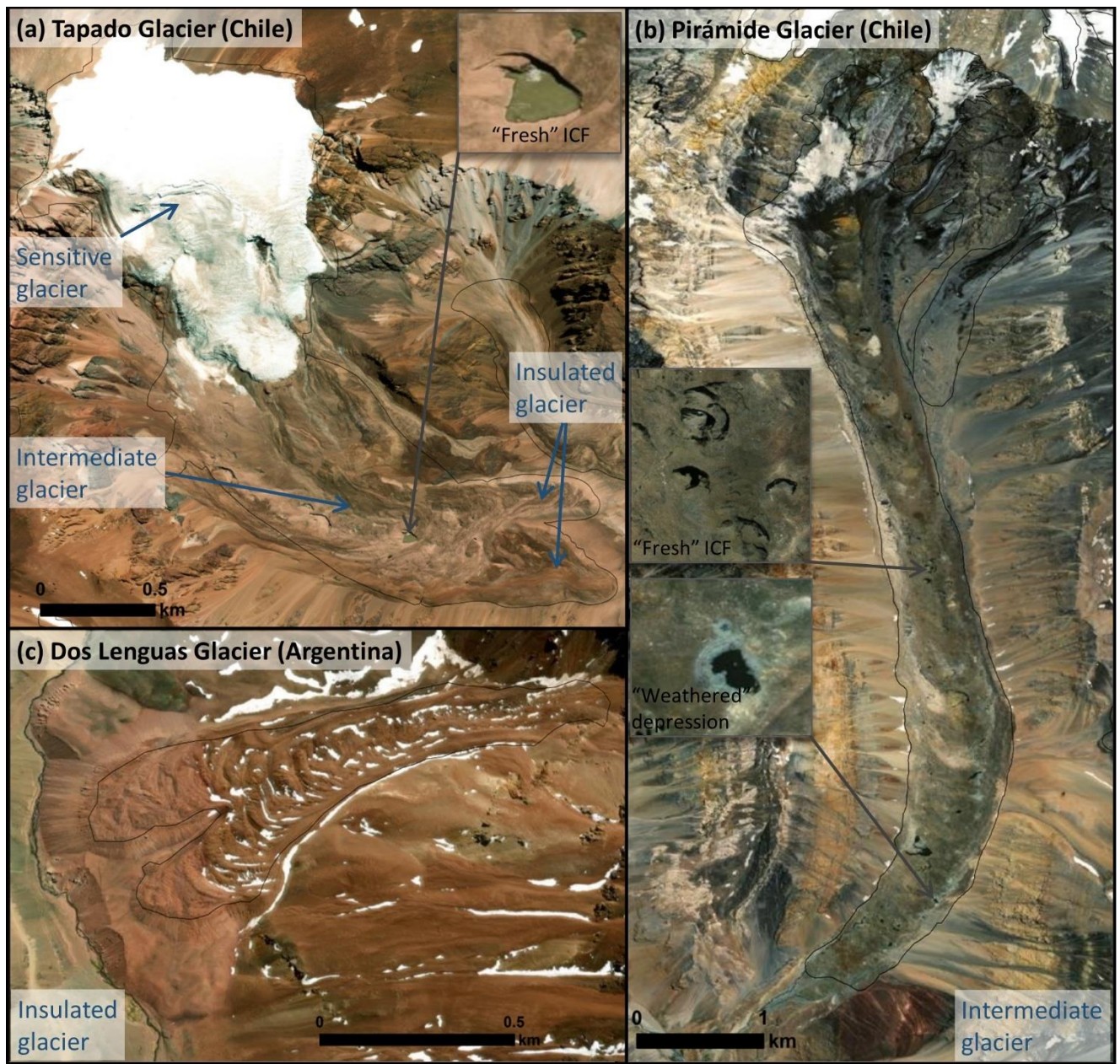

**Figure 1:** Three glaciers in the semiarid Andes for which the glacier-type (sensitive, intermediate or insulated) is clearly identifiable based on the geomorophological criterion presented in this paper are shown. Tapado Glacier is made up of the three distinct glacier types proposed in this study. Approximately 95 % of surface of Pirámide Glacier is covered by debris and there are numerous thermokarst depression features, so it is classified as an intermediate glacier. Dos Lenguas glacier does not have ice exposed at the surface, has convex topography accentuated with ridges and furrows, and an obvious frontal slope so it is classified as an insulated glacier. Image source (Esri basemap): (a) 11 March 2019 GeoEye (0.46 m), (b) 18 January 2013 WorldView-2 (0.5 m), (c) 17 September 2017 WorldView-2 (0.5 m).

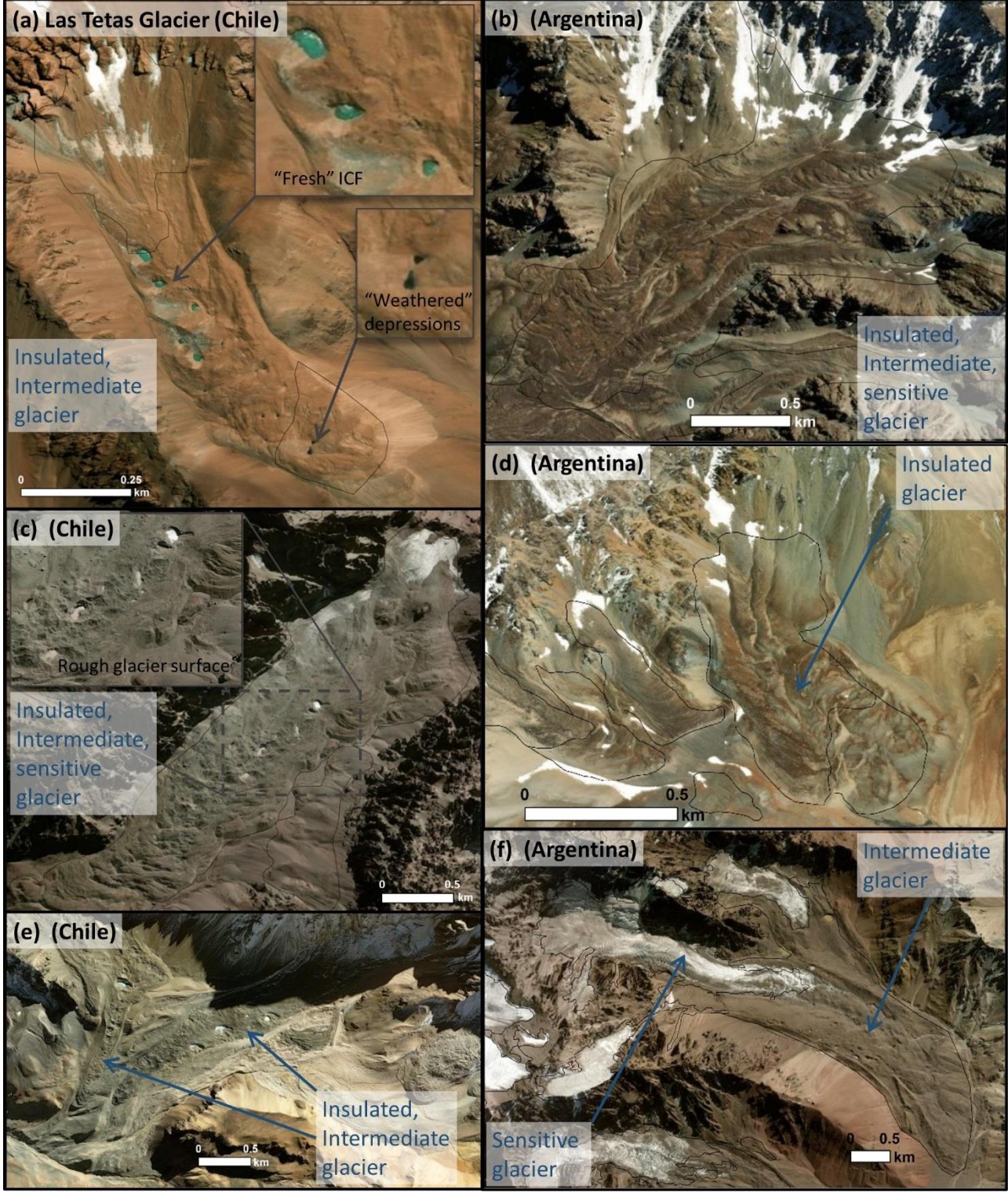

**Figure 2:** Examples from the semiarid Andes of Chile and Argentina are provided to clarify the proposed glacier types. All examples provided except for (d) contain multiple glacier types. (a) Las Tetas glacier is made up of an intermediate glacier in its upper portion, and an insulated glacier at lower elevations. (b) This glacier is dominated by the insulated glacier type while (c) and (e) are dominated by the intermediate glacier type and have a "rough glacier surface". (f) This glacier is a sensitive and intermediate glacier. (d) This glacier is a typical insulated glacier. Examples are provided of "fresh" ice collapse features (ICF) and "weathered" thermokarst depressions. The black outlines are glacier delineations from the national inventories. Image source (Esri basemap): (a) 11 March 2019 GeoEye (0.46 m), (b) 9 April 2018 GeoEye (0.46 m), (c) 1 April 2020 WorldView-2 (0.5 m), (d) 9 January 2018 WorldView-2 (0.5 m), (e) 6 May 2020 WorldView-2 (0.5 m), (f) 1 April 2020 WorldView-2 (0.5 m).

In the semiarid Andes of Chile the upward expansion of rock glacier morphology areas at the expense of debris-covered glaciers has been documented for two hybrid glaciers in the Colorado Valley (30° S) and Navarro Valley (33° S) that have debris-covered glacier morphology in their upper parts and rock glacier morphology in their lower parts (Monnier and Kinnard, 2017; Robson et al., 2022). In the Navarro valley a small debris-covered glacier has evolved into a rock glacier over the last half-century, and such transformations may result in glaciers being more resilient to changes in climate (Monnier and Kinnard, 2017). Other factors such as precipitation patterns may also change over time which can have an important influence on glacier mass balance (Burger et al., 2019) and water availability in general. These potential changes highlight the need for a glacier protection plan that is flexible and evolves through time.

**Table 1:** The category assigned for this article, the national inventories of Chile (DGA) and Argentina (IANIGLA), and in the published literature if available are listed for glaciers illustrated in the figures as well as for named glaciers in the published literature. There are two categories for this article: 1) All glacier types present and 2) the glacier type that we recommend for assigning a level of protection. The coordinates are provided for each along with the associated figure number or glacier name. A file with the location of each glacier in this table is included in the supplementary material (Glacier examples.kmz). The classifications for this study are based on the images displayed in the figures or using the images in the associated publication for glaciers not included in Figures 1 and 2. For classifications by Janke et al. (2015) there are three sub-classes within the class debris-covered glaciers: semicovered, fully covered, and buried glaciers.

| Figure ID / name | Latitude | Longitude | All categories present (this article) | Initial category for protection (this article) | Category DGA | Category IANIGLA | Published literature |
|---|---|---|---|---|---|---|---|
| 1a) Tapado | 30°9'15.42" S | 69°55'24.88" W | Sensitive/ intermediate/ insulated glacier | Sensitive glacier | Mountain glacier | -------- | Debris-free, debris-covered and rock glacier (Monnier et al., 2014, Pourrier et al., 2014) |
| 1b) Pirámide | 33°33'33.2" S | 69°53'35.1" W | Intermediate glacier | Sensitive glacier | Valley glacier | -------- | Debris-covered glacier (Ayala et al., 2016, Ferrando 2012), Fully covered (Janke et al., 2015) |
| 1c) Dos Lenguas | 30°14'42.26" S | 69°46'57.42" W | Insulated glacier | Insulated glacier | -------- | Rock glacier (active) | Rock glacier (Halla et al., 2020; Schrott, 1996) |
| 2a) Las Tetas | 30°10'9.00" S | 69°55'41.55" W | intermediate/ insulated glacier | Sensitive glacier | Mountain / rock glacier[a] | -------- | Debris-covered/rock glacier (Monnier and Kinnard, 2017) |
| 2b) | 33°39'25.69"S | 69°37'7.59"W | Sensitive/interm ediate/insulated glacier | Sensitive glacier | -------- | Debris-covered/ rock glacier | ------- |
| 2c) | 33°34'51.47" S | 70° 49.47" W | Sensitive/ intermediate/ insulated glacier | Sensitive glacier | Rock glacier | -------- | ------- |
| 2d) | 30°29'26.06"S | 70°10'29.90"W | Insulated glacier | Insulated glacier | -------- | Rock glacier (active) | ------- |
| 2e) | 34°13'7.04"S | 70° 60.80"W | Intermediate/ insulated glaciers | Sensitive glaciers | Rock glaciers | -------- | ------- |
| 2f) - west | 33° 9'53.46"S | 70° 2'14.97"W | Sensitive glacier | Sensitive glacier | -------- | Debris-free glacier | ------- |
| 2f) - east | 33°10'17.47"S | 70° 1'14.33"W | Intermediate glacier | Sensitive glacier | -------- | Debris-covered glacier | ------- |
| Juncal Norte | 32°59'42.8" S | 70°6'13.4" W | Sensitive glacier | Sensitive glacier | Valley glacier | -------- | Semicovered (Janke et al., 2015) |

| Glacier | Latitude | Longitude | | | | | |
|---|---|---|---|---|---|---|---|
| Llano de las Liebres | 30°14'44.49" S | 69°57'2.37" W | Insulated glacier | Insulated glacier | Rock glacier | ---------- | Rock glacier (Janke et al., 2015) |
| Navarro | 32°53'4.1" S | 70°2'31.1" W | Intermediate/ insulated glacier | Sensitive glacier | Mountain glacier / rock glacier[b] | ---------- | Semicovered/ fully covered/buried glacier/rock glacier (Janke et al., 2015; Monnier and Kinnard 2017) |
| Presenteser acae | 32°53'13.48" S | 70°1'44.87" W | Intermediate glacier[c] | sensitive glacier | Rock glacier | ---------- | Debris-covered/rock glacier (Monnier and Kinnard, 2017) |
| Tres Gomelos | 32°54'28.0" S | 70°1'36.3" W | Insulated glacier[d] | Insulated glacier | Rock glacier | ---------- | Rock glacier (Janke et al., 2015) |
| Universidad | 34°41'46.0" S | 70°19'55.5" W | Sensitive glacier | Sensitive glacier | Valley glacier | ---------- | Semicovered[e] (Janke et al., 2015) |
| El Paso | 30°13'58.43" S | 69°48'52.9" W | Insulated glacier | Insulated glacier | ---------- | Rock glacier (active) | Rock glacier (active; Croce and Milana, 2002) |

[a] The middle portion of this glacier containing many obvious "fresh" ice collapse features not included in the DGA inventory.

[b] Classifications are based on the glacier extent provided in Janke et al. (2015).

[c] While rock glacier morphology is present on the lower reaches of this glacier, the debris cover is not thick enough to insulate the glacier (>60 cm thick at lower elevations; Monnier and Kinnard 2017).

[d] This classification is based on the 2008 GeoEye-IKONOS imagery within Janke et al. (2015). However, more recent imagery (May 9 2020 World View, and Feb 10 2016 Google Earth) show several "fresh" ice collapse features that appear to have formed after 2008.

[e] Classification is based on our interpretation, not explicitly stated in the publication.

**Table 2:** Examples of intermediate glaciers with their initial category for protection (sensitive) and revised category for protection based on comparison with nearby glaciers using high-resolution datasets of glacier elevation change or modelled mass balance. A glacier ID is provided for each example along with a reference ID for glaciers it is compared to, both of which are points in the supplementary information file (category examples.kmz).

| Glacier ID | Latitude | Longitude | All categories present (this article) | Initial category for protection (this article) | Revised category for protection (this article) | Data source | ID of reference glaciers |
|---|---|---|---|---|---|---|---|
| 1.1 | 30°1′04.62″S | 69°55′44.21″W | intermediate/ insulated glacier | sensitive glacier | Intermediate glacier | Robson et al. (2022) | 1.2, 1.3 |
| 2.1 | 33°33′33.2″ S | 69°53′35.1″ W | Intermediate glacier | sensitive glacier | sensitive glacier | Ayala et al. (2016); Braun et al. (2019) | 2.2, 2.3 |
| 3.1 | 32°38′10.45″S | 70° 6′19.58″W | Intermediate glacier | sensitive glacier | sensitive glacier | Braun et al. (2019) | 3.2 |
| 4.1 | 34°12′57.12″S | 70° 6′10.19″W | Intermediate glacier | sensitive glacier | Intermediate glacier | Braun et al. (2019) | 4.2, 4.3 |

## 4 Examples from the semiarid Andes

Examples from the semiarid Andes of Chile and Argentina clearly illustrating the three glacier types as well as "fresh" ice collapse features and "weathered" depressions are shown in Fig. 1 and 2. Additional examples are included in Fig. 2 to help clarify. Details are provided in the figure captions and Table 1 summarizes the classification of each glacier in Figs. 1 and 2

according to the glacier categories proposed in this study, the categories defined by the Chilean and Argentinian national inventories, and within the published literature where references are available. Glaciers named and classified in the published literature have also been added to Table 1. The sensitive glaciers listed in Table 1 (the debris-free section of Tapado Glacier, Juncal Norte, Universidad glacier, and the sensitive glacier in Fig. 2f), the intermediate Pirámide Glacier and glaciers in Fig. 2e and 2f are included in the RGI. For all other hybrid glaciers only a small area at the highest elevation is included if ice is

exposed and insulated glaciers are excluded.

The most recent Chilean national inventory completed by the Dirección General de Aguas (DGA) defines rock glaciers as having no or almost no ice visible at the surface, generally convex topography, and a discernible frontal slope among other characteristics (DGA, personal communication, April 12 2021). It specifies that thermokarst features may be present but does

not indicate if these can be numerous or are rare. All other glacier types are categorized based on the Global Land Ice Measurement from Space (GLIMS) classification system (http://www.glims.org/MapsAndDocs/guides.html; DGA, personal communication, April 12 2021) which has two categories of interest for this discussion: 1) valley glaciers and 2) mountain glaciers, both of which include debris-free and debris-covered glaciers. Valley glaciers are generally confined to a valley whereas mountain glaciers are found on mountain slopes and include glaciers that do not fit into another category. There is no

differentiation with respect to the amount of debris cover. The most recent inventory is completed but not yet publicly available, so we have reviewed the preceding inventory which was used as a base for the revised inventory. Most glaciers classified as rock glaciers are insulated glaciers as defined in this study (similar to Fig. 1c, 2d). There are some glaciers with numerous "fresh" ice collapse features that have been categorized as rock glaciers (Fig. 2c, 2e). We suggest that when using the national inventory to evaluate water resources, that the categories proposed here additionally be applied to the area of interest so that

glaciers categorized as "rock glaciers" with numerous thermokarst depressions, especially "fresh" ice collapse features, can be differentiated from insulated glaciers since considerable mass loss may occur in the vicinity of these features (Ferguson and Vieli, 2020; Miles et al., 2016; Robson et al., 2022). Applying the proposed categories would also enable differentiation between sensitive and intermediate glaciers which could help facilitate the evaluation process.

Although rock glaciers are not explicitly defined in terms of the debris cover thickness in the Argentinian inventory completed by the Instituto Argentino de Nivología, Glaciología y Ciencias Ambientales (IANIGLA), the associated glacier inventory (https://www.argentina.gob.ar/ambiente/agua/glaciares/inventario-nacional) mostly agrees with the proposed categories. All glaciers classified as rock glaciers show no ice exposure, generally have convex topography and a discernible frontal slope

(e.g. Fig. 1c). There are many glaciers that have an upper portion that has "fresh" ice collapse features and/or is debris-free

and a lower portion characteristic of insulated glaciers (e.g. Fig. 2b, similar to Fig. 2a). These glaciers are characterized as debris-covered glaciers/rock glaciers which matches the classification we would propose here (intermediate/insulated glaciers). While far less common, there are some glaciers classified as rock glaciers that definitively have the characteristics of insulated glaciers except for having very large or numerous ice collapse features. We would like to suggest that these be labelled as intermediate/insulated glaciers (corresponds to debris-covered glaciers/rock glaciers in this inventory) for the purpose of water

resource evaluation. The category debris-covered glaciers in the Argentinian inventory is generally synonymous with intermediate glaciers as defined in this study evidenced by a near perfect match during a thorough review of the Argentinian inventory (supplementary material: Inventory area reviewed.kmz).

## 5 Discussion and concluding statements

We propose that glacier categories, used for the purpose of water resource evaluation and/or management, should reflect

differences in their sensitivity to environmental changes (e.g. temperature and precipitation). We suggest three categories: 1) Glaciers that are sensitive to environmental changes, 2) Intermediate glaciers, and 3) Glaciers that are thermally insulated from the environment.

Whilst there is inherent subjectivity in this proposal, we recommend that these categories are more appropriate for the purpose

of water resource evaluation and/or management than the available definitions based on glacier type in the scientific literature (Sect. 2) since these definitions can be more ambiguous than those proposed here and do not necessarily reflect the glacier's sensitivity. For example, a glacier that is almost fully covered with a thin layer of debris could be classified as a debris-covered glacier or as a rock glacier (e.g. Fig. 2e). Considering that such a glacier is more sensitive to changes in climate than an insulated one (e.g. Fig. 2d; Table 2, glacier 4.1) and the eastern portion is similar to Pirámide whose mass loss rate is

comparable to a debris-free glacier (e.g. Ayala et al., 2016), classifying it as a rock glacier could result in a false assumption that it is not very sensitive to environmental changes leading to an inappropriate level of protection.

The manual classification proposed in this study relies on individual interpretation of the geomorphology and is therefore somewhat subjective and limited. This simplified approach does not consider site-specific characteristics such as topography,

lithology, light-absorbing aerosols such as black carbon, or directly incorporate climate variables. We therefore propose that this be used as an initial classification which is later refined or used in combination with a more sophisticated and quantitative approach such as modelling the debris cover thickness and automating the classification by geomorphology using automatic detection methods. A global debris-cover thickness model only requiring input data that can be obtained remotely (geodetic mass balance and velocity fields) has been developed and these outputs could be used to help differentiate between sensitive

and intermediate glaciers (Rounce et al., 2021). However, outputs are limited to glaciers included in the RGI inventory and it

would be necessary to compare these outputs to measured debris thicknesses on glaciers in the semiarid Andes to evaluate their accuracy since the model was calibrated on a debris-covered glacier in Nepal. At present, methods for modelling thick debris cover (e.g. > 2 m) have not been validated so their effectiveness at differentiating between intermediate and insulated glaciers is unknown. The influence of debris cover on sensitivity could potentially be assessed in a more direct way since a relationship between satellite-derived surface temperatures and mass balance has been observed for debris-covered glaciers with debris thicknesses up to 0.4 m (Moore et al., 2019). Evaluation of the geomorphology and glacier delineation could potentially be completed in an objective way applying methods used to automatically detect debris-covered glaciers and rock glaciers (Lu et al., 2021; Robson et al., 2020). These approaches would allow for classification at a regional scale and could be used to identify individual glaciers where a more comprehensive analysis that accounts for precipitation, input from avalanches could be conducted. Temperature, precipitation, debris thickness, snow distribution and avalanche input could be modelled using a physically-oriented numerical model such as the TOPKAPI-ETH model which has already been applied successfully in the semiarid Andes region (Ayala et al., 2016). This type of model could also help identify tipping points (e.g. "peak water") which could provide very helpful information for policy decisions. This detailed modelling approach would require a large amount of input data (e.g. meteorological measurements from on and off glacier, glaciological measurements of mass balance, terrestrial photos, high resolution DEM, glacier outlines) which could only be obtained for select glaciers.

The classification proposed is specific to the semiarid Andes and is meant to function as an example upon which classification schemes for other regions could be based. The appropriate dividing line (debris thickness) between categories will vary from north to south along the Andes. For example, the study area between 29-34°S is characterized by cold and dry conditions which result in a glacier equilibrium-line altitude (ELA) that is generally several hundred meters above the 0°C isotherm (Masiokas et al., 2020) and short-wave radiation and sublimation are the primary melt processes (MacDonell et al., 2013; Réveillet et al., 2020). Further south in Patagonia (35°-55°S), most glaciers have their ELA below the 0°C, so rain may become an important factor influencing mass balance as seen in other regions (Wang et al., 2019), and the amount of incoming solar radiation is lower given the higher latitude. The former factor would likely increase the debris cover thickness required to impede persistent surface melt while the later would likely decrease the required thickness (Mattson et al., 1993). The distribution of dust and black carbon varies along the length of Chile (Rowe et al., 2019) and these particles have been modelled to reach glaciers at very high elevations such as the Tapado Glacier (> 4,500 m a.s.l.), but the impact of dust and black carbon on glacier mass balance within the study area is largely unknown (Rowe et al., 2019; Barraza et al., 2021). As glaciers within the study area are more sensitive to precipitation and albedo (short-wave radiation) compared to glaciers further south (e.g. Kinnard et al., 2020; MacDonell et al., 2013; Masiokas et al., 2020), they are likely more sensitive to impurities and a relatively thicker debris cover may be required here. As debris cover thickens the influence of local factors such as climate on glacier mass balance diminishes (Mattson et al., 1993), so the dividing line between sensitive and intermediate glaciers will likely vary more spatially than the dividing line between intermediate and insulated glaciers.

These categories are aligned with Janke et al. (2015) who propose six categories for debris-covered and rock glaciers. The categories in this paper additionally include debris-free glaciers and the number of categories have been reduced to three. Sensitive glaciers have experienced the highest mass loss rates in the La Laguna catchment (Robson et al., 2022) and this may be true elsewhere in the semiarid Andes. Insulated glaciers are expected to be less sensitive and provide longer-term reservoirs (Jones et al., 2018) and are expected to become increasingly important in a warming climate as the contribution from more sensitive glaciers diminishes (Ferguson and Vieli, 2020; Jones et al., 2018). It is likely that they also play a role storing and delaying runoff (Winkler et al., 2016). Their value as a water resources is region-specific, with a more significant role in areas that are water-scarce and rock glaciers are the dominant glacier type such as the semiarid Andes (Azócar and Brenning, 2010; Jones et al., 2018; Schaffer et al., 2019). Here, an elevated level of protection may be needed, focusing protection on individual glaciers may not be sufficient and will likely need to be expanded over larger regions to capture the sum of water reserves contained within rock glaciers and other ice-rich landforms to meet the needs of society. The Chilean and Argentinean GPL do not identify the distinct role glacier types provide in terms of water resources as described above. The GPL also do not consider water availability and how this varies with latitude and with time. If these were incorporated into legislation, it would be possible to match the level of protection to the need resulting in protection that would be region-specific, meet the needs of society without over- or under-protecting, and could evolve through time as the climate and water availability changes. Water availability could be coarsely identified with the water-scarcity levels identified for all regions in Chile within the national Atlas Del Agua and national water plan (Plan Nacional del Agua) for Argentina.

The specific decisions with regards to the level of protection for each region and assigned to each glacier category proposed here are public policy decisions that require balancing many factors such as water resources and the economy and are beyond the scope of this paper. To support informed decision making with respect to the protection of glaciers we suggest that information on the sensitivity and hydrological value of different glacier types be explicitly provided in an easily accessible way, particularly for regions that are expected to be water-scarce in the coming decades as longer-term water reservoirs may be of critical importance. In general, we suggest that the level of protection match the needs of society as a minimum, ideally stringent enough to also sustain biodiversity, sustainable tourism, traditional practices from indigenous communities and scientific investigation in key areas. A conservative approach should be taken given that the semiarid Andes region is already water-scarce (29-34°S) and there is currently insufficient data to evaluate the current or future hydrological contribution to streamflow from rock glaciers and ice-rich ground (Schaffer et al., 2019; Mathys et al., submitted).

The number of categories has been reduced to the minimum needed to distinguish glaciers by their sensitivity to changes in the environment (three categories) to facilitate relatively easy and efficient identification of the glacier types while retaining sufficient detail to designate an appropriate level of protection and monitoring protocol associated with the GPL and EIA processes. Both the Chilean and Argentinian inventories mostly agree with the division between intermediate and insulated

glaciers. The only exception is for glaciers categorized as rock glaciers that also have thermokarst depressions, particularly "fresh" ice collapse features. We would like to suggest that for the purpose of water resource evaluation these be categorized as intermediate/insulated glaciers in general and considered intermediate glaciers for evaluating the level of protection since considerable mass loss may occur in the vicinity of these features (Ferguson and Vieli, 2020; Miles et al., 2016; Robson et al., 2022). The Argentinian national inventory effectively differentiates between sensitive and intermediate glaciers for the focus area (~27°-35° S), while the Chilean inventory does not. We would suggest adding this distinction when classifying glaciers for the purpose of water resource evaluation in Chile. We hope that these suggestions and the classification scheme proposed will be useful for public policy, as a complement to the generalized guidelines for glacier protection outlined in the GPL for Argentina and Chile, possibly to improve the current Chilean EIA which treats all glacier types as one category and for monitoring. We envision the methodology outlined in this paper as an initial classification that could be efficiently completed at a national scale and added as a layer to the existing national inventories, potentially by glaciology professionals who created the national inventories (DGA in Chile, IANIGLA in Argentina), using data already available (e.g. high resolution satellite imagery). A more sophisticated and quantitative approach could be applied as the data and advancements in methodology required become available. However, this approach would require much more time, expert professionals and in situ data, so it may be challenging given that there are no trained glacier professionals in the EIA system or local government departments in Chile.  In addition to their hydrological value we also recommend other values such as ecosystem services provided by glaciers, their scientific importance, potential for sustainable tourism, importance for cultural and natural heritage, presence in a protected area (not limited to national parks), and the rights of indigenous communities be considered within the evaluation process, with the level of protection elevated for glaciers providing these additional benefits to society.

**Author contribution and competing interests**

Nicole Schaffer prepared the manuscript in collaboration with Shelley MacDonell who contributed content and helped write the manuscript. Whilst the authors declare they have no conflict of interest, we acknowledge that Shelley MacDonell has participated in working groups and panels with respect to the creation of glacier protection legislation in Chile, has conducted training courses for members of the Environmental Impact Assessment system and acted as a reviewer for the Chilean National Glacier Inventory that is currently being finalised.

**Acknowledgements**

This work was supported by ANID + Concurso de Fortalecimiento al Desarrollo Científico de Centros Regionales 2020-R20F0008-CEAZA, FIC-R (2016) Coquimbo (BIP: 40000343), and ANID-CENTROS REGIONALES R20F0008. Nicole Schaffer was supported by CONICYT + FONDECYT + Postdoctorado (3180417). In addition, we would like to acknowledge

conversations with researchers, professionals and others in the Chilean and Argentinian communities, including the Dirección

General de Aguas, for discussions related to glacier protection, the Environmental Impact Assessment process and glacier behaviour in general that both inspired and informed this work. Additionally, fruitful conversations with members of the 'International Permafrost Association action group on rock glacier inventories and kinematics' with respect to rock glacier inventorying are gratefully acknowledged.

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
