# Peer review of "Brief communication: A framework to classify glaciers for water resource evaluation and management in the Southern Andes"

_The Cryosphere, 2021_

## Author Comment (AC1)

**Reviewer 1**

Review of "Brief communication: A framework to classify glaciers for water resource evaluation and management in the Southern Andes" by Shaffer & MacDonnell

**Summary**

This brief communication proposes a new classification of the glacier landforms present in the Andes. The classification focuses on the sensitivity of the landforms to climate change and their hydrological impacts rather than purely on their geomorphological traits. The proposed classification is suggested to contribute to the discussion on the development of glacier protection legislation in both Chile and Argentina, which up to now have been relatively unsuccessful.

**General comments**

In general, the manuscript is well written and the message the authors intend to convey is clear. However, I have to say I am slightly confused about the intention of this communication. On the one hand, I do see the benefit of publishing this work in The Cryosphere, as this discussion may also exist in other parts of the world and a consensus in identifying glacier sensitivity from a policy standpoint could be beneficial. To serve this purpose, I do think the manuscript in its present form is (too) much focused on the Andean case. On the other hand, I wonder whether (the message of) this manuscript wouldn't be a better fit for a journal or other medium that allows direct targeting of the intended audience, i.e. policy makers, nature conservatists and water resource managers in the respective countries. I am not saying I do not see the benefit for TC and a "general" audience, but a more general focus would better support that.

RESPONSE: We decided to submit this article to The Cryosphere because we thought it would be very valuable to receive feedback from other glaciologists around the world to help build a consensus within the scientific community on identifying glaciers based on sensitivity for the purpose of water resource policy and management. To meet this objective, it was necessary to include technical details and information very specific to glaciology. Once this initial objective is met, a secondary document could be written that is less technical aimed specifically at policy makers, water resource managers and the general public.

We focus the paper on the semiarid Andes as a case study since this area is particularly relevant for water resource evaluation, legislation and management given that it is water-scarce, a large portion of glaciers are found outside of national parks, and it has a relatively high population density. Such high mountain areas are expected to see the largest temperature increases by the end of the twenty-first century. Areas of particular concern are in Ecuador, Peru, Bolivia, and northern Chile (Bradley et al. 2006; Souvignet et al. 2010; MRI Working Group 2015). Therefore, we think that a case study on the semiarid Andes is highly relevant within a global context. We decided to narrow our focus to a relatively small region so that we could provide very specific and concrete guidelines for defining the glacier categories. Given the large variability along the Andes in climate, topography and glacier characteristics, the appropriate debris thickness threshold among other criteria differentiating the categories will vary from north

to south so broadening the paper scope to encompass all of Chile and Argentina or a larger area would have required very general guidelines which we think would be less useful. Our aim is that the classification proposed for the semiarid Andes can serve as an example upon which classification schemes for other regions in Chile or other high mountain areas could be based.

I am also wondering how relevant it really is to identify the different landform types from a legislation perspective. Apparently, the political discourse has not yet been fruitful with respect to the GPL, even when just considering them as a single entity. Wouldn't introducing a system of sensitivities complicate things even further? In my opinion, the current manuscript does not express clearly enough how the introduction of the proposed classification would benefit the discourse around glacier protection, how it would benefit drafting related legislation, and how water resource management will be improved as a result.

RESPONSE: After careful consideration, we agree that introducing the proposed classification would likely complicate the proposed GPL and make it more difficult to pass this law. However, the currently proposed GPL is limited in its ability to effectively protect glaciers as a single classification for all glaciers makes it rigid in both space and time.

The proposed classification would benefit the discourse around glacier protection by initiating a discussion on the distinct contribution different glacier types can make toward helping to meet water-resource needs, particularly over decadal or longer time scales. For example, glaciers that are more sensitive to changes in climate (e.g. debris-free glaciers) provide a relatively large annual contribution to streamflow now, while rock glaciers are less sensitive and provide a longer-term reservoir (Jones et al., 2018). Sensitive glaciers are more responsive to climatic changes and in the Southern Andes (south of ~25° S) the vast majority of glaciers have already reached or are expected to reach their maximum runoff or "peak water" before 2050 with a decrease in runoff thereafter (Burger et al., 2019; Huss and Hock, 2018). Therefore, in the coming decades insulated landforms will become increasingly important.

Classifying different glacier types in a way that reflects their distinct hydrological roles opens the possibility for more flexible legislation that can match the level of protection to the need resulting in protection that would be region-specific, meet the needs of society without over- or under-protecting, and could evolve through time as the climate and water availability changes. The current law is likely to under-protect in water-scarce regions such as from Santiago (Chile) to the north, assuming it only includes active rock glaciers, and may over-protect in areas with abundant water reserves (e.g. Patagonia) potentially limiting economic activity that could reasonably be carried out with precautions given that water from rock glaciers here is not likely critical. If the level of protection was linked to water-scarcity levels by region, the level of protection could be modified as water-scarcity levels change through time.

Water resource management would be improved with these classifications. Currently, many of the requirements in the EIA process are the same regardless of glacier type, and variable impacts are not given adequate consideration. For example, a debris-free glacier would be more sensitive to air particles such as black carbon from a nearby road than a debris-covered glacier, but this difference cannot be adequately addressed within the current EIA. Monitoring requirements would be more relevant if they were glacier-type specific. For example, obtaining an ice core

from a debris-free or debris-covered glacier is relatively straight forward and requires equipment that can be carried on foot, while obtaining one from a rock glacier is difficult and requires a much more robust setup that is difficult to transport. The most effective method to measure mass balance also differs between glacier types.

We have modified the first paragraph of the introduction to focus less on GPL and more on the general benefit of these classifications for legislation and the EIA as outlined above. We have also added some additional text on line 257 of the discussion suggesting that the level of protection could be matched to water needs.

I agree that the (quite minor) redefinition in classes defined by the authors with respect to traditional geomorphological categories of clean-ice, debris-covered and rock glaciers could improve assessment in terms of sensitivity and hydrological impacts up to a certain extent. However, in essence, the classification is still just based on a simple interpretation of the surface morphology, which is an oversimplification. This results in the straightforward and broad classes "sensitive" vs "insensitive", which may be too much of a black and white approach to be really useful in practice. High heterogeneity and variability exist among glaciers in their sensitivity and hydrological response, and this is for a considerable part irrespective of glacier surface type. It may be due to other geomorphological specifics of a glacier that are not considered in the proposed classification (e.g. slope, elevation, bed lithology, aspect etc.), but also due to differences in local climate, local anthropogenic disturbances, and possible feedbacks therein. Could some of these components be included somehow? Wouldn't an (even simple) modelling approach allow for a more informative estimation of the actual sensitivity of the glaciers? I would suggest the authors to at least elaborate on the limitations of such a simple classification and place it into a context of other, more developed approaches such as regional and/or individual glacier modelling. "Advanced" approaches would also be better to identify potential tipping points and transient effects, which could be very important arguments in policymaker discussions and conveying the urgency of expected changes in hydrology.

RESPONSE: We agree that the qualitative approach proposed here is simplistic compared to the heterogeneity and variability that exist among glaciers. We envision the methodology outlined in this paper as an initial classification that could be efficiently completed at a national scale using data already available (e.g. high resolution satellite imagery). In the paper we now suggest that a more sophisticated and quantitative approach that could consider topography, climate, anthropogenic factors such as black carbon be applied as the data, advancements in methodology required, and qualified personnel become available. However, this approach would require much more time, expert professionals and in situ data, so it may be challenging given that there are no trained glacier professionals in the EIA system or local government departments in Chile and there is very limited in situ data available to complete a more sophisticated and quantitative modelling approach at a regional scale. We have modified the discussion paragraph starting on line 227 to suggest this two-tiered approach (an initial classification as outlined in this paper, followed by a more quantitative and sophisticated approach). We have also modified and expanded upon the quantitative approaches suggested. We also state that using such physically-oriented numerical models to identify tipping points (e.g. "peak water") could provide very helpful information for policy decisions. Finally, we have explicitly identified the limitations of the quantitative approach presented in this paper at the beginning of this paragraph (line 228).

We have added a paragraph at line 49 discussing the large variation in climate, topography, and glacier characteristics that exists from north to south in the Andes and recognize that the dividing line (debris thickness threshold between categories) will vary from north to south. We clarify here and, in the discussion, that the study area chosen is meant to function as an example upon which classification schemes for other regions could be based. We have added a new paragraph starting on line 243 that details how the dividing line might vary from north to south and why.

A simple modeling approach could be applied such as a temperature-index model that includes solar radiation. However, above 4000 m a.s.l. the performance of temperature-index models is poor within the study area (Ayala et al., 2017). Additionally, this type of model would not be able to incorporate debris thickness and would therefore not provide realistic results for sensitivity. A debris-cover model would need to be used to calculate the thickness, then this would need to be incorporated into a mass balance model capable of accounting for debris-cover. A global debris-cover thickness model only requiring input data that can be obtained remotely (geodetic mass balance and velocity fields) has been developed and these outputs could be used to help differentiate between sensitive and semi-sensitive landforms (Rounce et al., 2021). The outputs from an earlier version of this model compare well to measurements of debris thickness on Pirámide Glacier (Ayala et al., 2016), but comparison with other glaciers in the semiarid Andes is necessary to evaluate the accuracy since the model was calibrated on a debris-covered glacier in Nepal. At present, methods for modelling thick debris cover (e.g. > 2 m) have not been validated and are therefore not a reliable tool to differentiate between semi-sensitive and insulated landforms.

I do not really understand the difference between landform and glacier used in the manuscript. A glacier seems to me as single entity, especially since it is hydrologically connected, but here it is suggested that a glacier is actually a landform that can consist of multiple glacier types. I would suggest using a better description of and distinction between these terms.

RESPONSE: We agree and have changed all instances of "landform" to "glacier" in the manuscript.

Also, taking the most sensitive part from a geomorphological perspective in a "hydrologically-connected" case to represent the sensitivity of the entire glacier/landform is not necessarily valid. The system should rather be classified as a whole. This goes back to my previous point: will this simple classification adequately represent sensitivity of the existing wide range of glaciers and glacier systems? For all glaciers, but particularly for a multi-type ones, sensitivity very much depends on the type of external forcing that causes a potential disturbance. If, for instance, the lower part of a glacier system is heavily debris-covered, it could be relatively insensitive to climate warming due to the insulation the debris provides in the ablation zone, but could be highly sensitive to processes that affect accumulation zone albedo such as the snowfall frequency or black carbon deposits.

RESPONSE: We have added a sentence on line 123 to indicate that assigning a level of protection associated with the most sensitive category is an initially conservative approach. When more information becomes available, the sensitivity level can be downgraded if justified.

We agree that the qualitative approach proposed here is simplistic compared to the heterogeneity and variability that exist among glaciers. We envision the methodology outlined in this paper as an initial classification that could be efficiently completed at a national scale using data already available (e.g. high resolution satellite imagery). In the paper we now suggest that a more sophisticated and quantitative approach that could consider topography, climate, anthropogenic factors such as black carbon be applied as the data, advancements in methodology required, and qualified personnel become available. However, this approach would require much more time, expert professionals and in situ data, so it may be challenging given that there are no trained glacier professionals in the EIA system or local government departments in Chile and there is very limited in situ data available to complete a more sophisticated and quantitative modelling approach at a regional scale. We have modified the discussion paragraph starting on line 227 to suggest this two-tiered approach (an initial classification as outlined in this paper, followed by a more quantitative and sophisticated approach). We have also modified and expanded upon the quantitative approaches suggested. We also state that using such physically-oriented numerical models help identify tipping points (e.g. "peak water") which could provide very helpful information for policy decisions. Finally, we have explicitly identified the limitations of the quantitative approach presented in this paper at the beginning of this paragraph (line 228).

We have added a paragraph at line 49 discussing the large variation in climate, topography, and glacier characteristics that exists from north to south in the Andes and recognize that the dividing line (debris thickness threshold between categories) will vary from north to south. We clarify here and, in the discussion, that the study area chosen is meant to function as an example upon which classification schemes for other regions could be based. We have added a new paragraph starting on line 243 that details how the dividing line might vary from north to south and why.

We agree that the sensitivity could change along the length of the glacier (ablation versus accumulation area). However, we expect that the variability in sensitivity within a given class (e.g. ablation versus accumulation area of an insulated glacier) will be less than the variability in sensitivity between classes. Since our aim is to propose a classification system that can differentiate between classes, we think that the proposed scheme is sufficient as an initial classification, which can later be modified using more sophisticated methods as described in the discussion.

SPECIFIC COMMENTS

L159-166. This is a good point, but it further reveals the complications with the sensitivity classification. I agree that a protection plan should evolve over time, but it should ideally already account for these temporal processes and effects from the get go. Is there any way temporal evolution could be included in the sensitivity classification approach? How would this affect the discourse and development of GPL?

RESPONSE: Yes, the temporal evolution could be included in the sensitivity classification approach. Since this approach distinguishes between different glacier types, it would be possible to renew the classification of glaciers every 10 years or so and in that way incorporate changes in glacier type. If for example a debris-covered glacier has evolved into a rock glacier over time, the classification would change and this would potentially impact the level of protection assigned

and hydrological role associated with that landform (e.g. short-term contributor to streamflow versus long-term reservoir). This would help to facilitate a discussion on and offer an opportunity to incorporate glacier evolution over time and the associated changes in streamflow contribution and hydrological role into the GPL.

L261-263. I am not sure why it is necessary or even desirable that non-experts can determine the sensitivity of a glacier. A well-developed database of glacier sensitivities created by experts using thorough analysis will deliver a much more insightful indication of the sensitivity of glaciers in a region or catchment and will serve policymakers better.

RESPONSE: We agree. Initially we proposed that the classification could be done by non-experts since there are no trained glacier professionals in the EIA system or local government departments in Chile. However, the Dirección General de Aguas (DGA, Chilean Water Authority, from the Ministry of Public Works) has a small unit that focuses on snow and ice that could complete a database or a glaciologist with knowledge of glaciers in Chile/Argentina could be contracted to do this. We have added a sentence on line 274 stating that the initial classification proposed in this manuscript could potentially be completed at a national scale by glaciology professionals who created the national inventories (DGA in Chile, IANIGLA in Argentina).

**References:**

Ayala, Alvaro, Pellicciotti, Francesca, MacDonell, Shelley, McPhee, James and Burlando, P. (2017) Patterns of glacier ablation across North-Central Chile: Identifying the limits of empirical melt models under sublimation-favorable conditions. Water Resources Research, 53 (7). pp. 5601-5625. ISSN 0043-1397

Ayala, A., Pellicciotti, F., MacDonell, S., McPhee, J., Vivero, S., Campos, C. and Egli, P.: Modelling the hydrological response of debris-free and debris-covered glaciers to present climatic conditions in the semiarid Andes of central Chile, Hydrol. Process., 30(22), 4036–4058, doi:10.1002/hyp.10971, 2016.

Bradley RS, Vuille M, Diaz HF, Vergara W (2006) Threats to water supplies in the tropical Andes. Science 312(5781):1755–1756. https://doi.org/10.1126/science.1128087

Burger, F., Ayala, A., Farias, D., Shaw, T. E., MacDonell, S., Brock, B., ... & Pellicciotti, F.: Interannual variability in glacier contribution to runoff from a high-elevation Andean catchment: understanding the role of debris cover in glacier hydrology. Hydrological Processes, 33(2), 214-229, 2019.

De Pasquale, G., Valois, R., Schaffer, N. and MacDonell, S.: Geophysical signature of two contrasting Andean rock glaciers, Crosph., submitted.

Huss, M. and Hock, R.: Global-scale hydrological response to future glacier mass loss, Nat. Clim. Chang., 8(2), 135–140, 325 doi:10.1038/s41558-017-0049-x, 2018.

Jones, D. B., Harrison, S., Anderson, K. and Betts, R. A.: Mountain rock glaciers contain globally significant water stores, Sci. Rep., 8(2834), doi:10.1038/s41598-018-21244-w, 2018.

MRI Working Group (2015) Elevation-dependent warming in mountain regions of the world. Nat Clim Chang 5:424–430. https://doi.org/10.1038/nclimate2563

Rounce, D. R., Hock, R., McNabb, R. W., Millan, R., Sommer, C., Braun, M. H., Malz, P., Maussion, F., Mouginot, J., Seehaus, T. C. and Shean, D. E.: Distributed Global Debris Thickness Estimates Reveal Debris Significantly Impacts Glacier Mass Balance, Geophys. Res. Lett., 48(8), e2020GL091311, doi:10.1029/2020GL091311, 2021.

Souvignet M, Gaese H, Ribbe L, Kretschmer N, Oyarzún R (2010) Statistical downscaling of precipitation and temperature in northcentral Chile: an assessment of possible climate change impacts in an arid Andean watershed. Hydrol Sci J 55(1):41–57. https://doi.org/10.1080/02626660903526045

Schaffer, N., MacDonell, S., Réveillet, M., Yáñez, E. and Valois, R.: Rock glaciers as a water resource in a

changing climate in the semiarid Chilean Andes, Reg. Environ. Chang., 19, 1263–1279, doi:10.1007/s10113-018-01459-3, 2019.

---

## Author Comment (AC2)

**Reviewer 2**

**General comments:**

1) The manuscript is well written and presents a sound classification schema of glaciers based on their sensitivity to environmental change. The methodology has solid support in the literature, and the consideration of this new classification is likely to be a valuable contribution to the management of glaciers, especially concerning their hydrological services.

2) The work is presented as a contribution towards the development of GPL, particularly as a solution to inadequate definitions in the Chilean GPL projects. In that sense, there is no clear articulation between the proposed classification and the GPLs. Both Chilean and Argentinean GPLs avoid conflict and ambiguities by protecting all glaciers equally, regardless of their type, size, location or debris cover. In that context, it is hard to understand how this classification schema can help the design of a law proposal of consensus and without the "legal issues" mentioned for the Argentinean GPL. If the authors propose a type-dependent level of protection (as stated in lines 224-225, 261), that should be clearly stated and followed with well-elaborated reasoning to support that proposal. Arguably, a type-dependent level of protection will only complicate things, especially given that the classification is sometimes ambiguous (lines 115-116) and changes with time (lines 159-164). In some sections, it even seems that the authors suggest a case-by-case assignation of the level of protection (lines 274-278). Much emphasis was put on the usefulness of the proposed classification for glacier management. However, the Argentinean GPL and Chilean GPL proposals aim to protect glaciers, not to manage them. If the authors are mainly suggesting a type-dependent monitoring program or the addition of this classification to national inventory fields (as stated in line 271), this should be clearly stated from the start. In such a case, they should also include a more detailed explanation of how this classification will help water resources management and a throughout motivation of the methodology. For example, why a classification is better than a "sensitivity index" or case-by-case modelling.

RESPONSE: We have modified the first paragraph in the introduction to highlight the usefulness of the classification scheme proposed with respect to legislation in general, including the GPL. We have also added a few sentences to the discussion starting on line 257 to highlight how incorporating the classification scheme in combination with water-scarcity could improve the GPL by making it possible to match the level of protection to the water resource need resulting in protection that would be region-specific, meet the needs of society without over- or under-protecting, and could evolve through time as the climate and water availability changes. We agree that introducing the proposed classification would likely complicate the proposed GPL and make it more difficult to pass this law. However, the currently proposed GPL is limited in its ability to effectively protect glaciers as a single classification for all glaciers makes it rigid in both space and time. We have also added a couple sentences to the first paragraph of the introduction to highlight the usefulness of the proposed classification for glacier management (EIA).

We are proposing a type-dependant level of protection. We have added a paragraph to the discussion starting on line 260 that addresses the level of protection. We think that the specific

decisions with regards to the level of protection for each region and assigned to each glacier category proposed are public policy decisions that require balancing many factors such as water resources and the economy and are beyond the capacity of authors of this paper to decide. However, we do provide some general recommendations.

In the discussion we have added a sentence to suggest that the classification scheme be added to the national inventory by glaciology professionals who created the national inventories (DGA in Chile, IANIGLA in Argentina). We have added information to the paragraph starting on line 227 of the discussion on the limitations of case-by-case modelling (lots of in situ data required) and advantages of applying the classification scheme (can be completed efficiently with data already available).

3)Following the facts detailed in lines 31-32, it seems inaccurate to refer to glaciers/landforms as sensitive/insensitive. The differences seem to be related only to the timescales of their response to environmental changes. Maybe fast/slow response might be better terminology.

RESPONSE: We agree that in the submitted version of the paper the differences are related only to timescales in their response to environmental changes. Our initial idea was a broader scope that also includes factors such as sensitivity to light-absorbing aerosols (e.g. black carbon). We have incorporated some text on black carbon into the manuscript introduction, section 3 and the discussion so that this is clear. Given that we have clarified the scope of "environmental changes" we have continued to use the terms "sensitive/insulated".

4) The main proxy to assess glacier change is mass balance, which depends on accumulation and ablation. However, this works seems to focus entirely on the ablation part of the equation.

In avalanche-fed glaciers, which is often the case for categories 2 or 3 (semi-sensitive and insulated), there could be a high climatic sensitivity associated with the snow accumulation on surrounding slopes that are not even part of the glacier. While such glaciers would "melt away slowly" due to their debris cover, their mass gain mechanism might have a very high sensitivity to environmental changes. In these cases, their water storage capability at inter-annual timescales would also have a high sensitivity to environmental change.

RESPONSE: It is true that debris-covered glaciers that are avalanche-fed are sensitive to changes in precipitation (Burger et al., 2019). However, we expect that an avalanche-fed semi-sensitive glacier would still be less sensitive to climate than a sensitive glacier. Ayala et al. (2016) found this to be the case for Pirámide glacier in the semiarid Andes. This mainly avalanche-fed debris-covered glacier was considerably less sensitive to changes in climate than the other two nearby debris-free glaciers studied. Our aim is to incorporate factors that will help distinguish between the categories, rather than factors that result in variable sensitivity within a category, so we have decided not to include the sensitivity associated with avalanches when initially defining the categories with optical imagery. In the discussion we do suggest that the initial classification could be refined using various approaches including physically-oriented numerical mass balance modelling within which the impact of avalanche input could be incorporated.

5) The use of the term "landform" makes the manuscript very confusing. While it can refer to anything (a glacier, a ridge, a mountain), it is often used to refer to a glacier, where the direct use of the term "glacier" would make the text much clearer. In some cases, for the same glacier the text says that it is a landform composed of multiple glacier types, and that it is a glacier composed of multiple landform types (line 118: "Where a landform is made up of multiple glacier types (Fig. 1a [Tapado Glacier])", lines 125-126 "Tapado Glacier [Fig. 1a] is made up of the three distinct landform types..."). Other sections use the concept of "glacier morphology" (line 161). More consistent use of the terminology is necessary: "Glacier" and "surface-type" could be better concepts to use (instead of randomly interchange either of those by "landform").

RESPONSE: We agree and have changed all instances of "landform" to "glacier."

6) In the context of GPL and glacier inventories. It seems that the authors propose the use of their methodology nationwide or throughout the Andes. However, the examples presented in figures and Table 1 are biased to the semi-arid Andes; the same is true regarding the accuracy check proposed in line 234. All examples are within four degrees of latitude. It must be clear what is the geographical area for which this methodology has been designed. If the application area is the whole of the Andes, the authors should address the different challenges posed by tropical and Patagonian glaciers.

RESPONSE: We have added a paragraph to the introduction starting on line 49 to explain why we chose to focus on the semiarid Andes (this area is particularly relevant for water resource evaluation) and clarified that the semiarid Andes scheme provided is meant to serve as an example upon which classification schemes for other regions could be based. We have also added a paragraph to the introduction discussing the large variation in climate, topography, and glacier characteristics that exists from north to south in the Andes and recognize that the dividing line (debris thickness threshold between categories) will vary from north to south. We have added a new paragraph starting on line 243 that details how the dividing line might vary from north to south and why with an emphasis on the difference between the semiarid Andes and Patagonia.

**Specific comments (numbers refer to manuscript version 2) :**

7: In the context of this paragraph and in particular the GPLs, "landform types" have a very different and more specific meaning than used in the rest of the text, as the most controversial definitions that have hindered consensus of the Chilean GPL are the definitions of Glacier, Periglacial, and permafrost. However, "landform types" in the manuscript refers interchangeably to glaciers or parts of a glacier with a distinct surface type (based on debris cover). This difference gives the impression to the reader that this work offers a direct solution to the definitions controversy that, has been in part, the cause of the lack of consensus, which is wrong.

RESPONSE: We agree that the classification scheme proposed does not directly resolve the definitions controversy that has hindered the consensus of the Chilean GPL. We have modified the introduction to present the usefulness of the classification scheme in a broader context to help effectively protect, manage, and monitor glacier water resources by differentiating between glacier types.

21-22: Given that the authors seem to be opening the discussion over the idea of not protecting all glaciers equally but depending on their hydrological behaviour. It seems very important to elaborate on what legal issues have hindered the application of the Argentinean GPL, or at least give a reference for that affirmation.

RESPONSE: We have broadened the scope of the introduction to present the usefulness of the classification scheme for legislation and the EIA. Given the reduced focus on GPL we do not think it is necessary to elaborate on the legal issues that have hindered the application of the Argentinian GPL.

23-24: This requires further elaboration. It is unclear how distinguishing between glacier types can reduce the legal ambiguity. In general, one would think that the current approach of Chilean and Argentinean GPLs (protecting all glaciers regardless of type) is less ambiguous than differential protection based on a glacier classification schema.

RESPONSE: Defining the different glacier types included in the GPL would clarify which glacier types are in fact protected. For example, it is currently not clear if rock glaciers are protected or not. Some rock glaciers (primarily active rock glaciers) are included in the national inventory so one could assume that these are protected, but this is not explicitly indicated in the law. Assuming rock glaciers are included, a practical definition of rock glaciers should be included in the GPL to clarify which landforms are considered rock glaciers (e.g. only active rock glaciers or also inactive ones?). The definition provided in this paper for insulated landforms could be used for this purpose.

39: The switch between the "glacier" terminology and the use of "landform" should be explained here. Otherwise, simply keep using "glacier."

RESPONSE: We agree and have changed all instances of "landform" to "glacier."

77-80: It seems against the objectives of this work to base the threshold of debris thickness on a single glacier. Arguably, debris type can have a significant influence, as well as the partitioning of the different melt processes affecting a glacier. In areas where sublimation is the primary melt process, a thin layer of debris might be enough to reduce melting significantly. In other cases, such as the temperate glaciers of New Zealand and Patagonia, a large amount of the melting is due to rain, and perhaps a much thicker debris cover is required to reduce melt rates. Pirámide glacier might be representative only of glaciers where shortwave radiation is the dominant melting process.

RESPONSE: We agree and have therefore included an additional paragraph in the discussion to address how the threshold of debris thickness might change with latitude (starts on line 243).

121-123: Again, it seems against the objectives of this work to include ambiguous criteria like this (what is "very minor"?).

RESPONSE: We agree and have specified what we mean by "very minor" ($< \sim 20\%$ of the surface area).

Figure 2: Please include coordinates or some ID (either in the figure or caption) for all unnamed glaciers (b-f). Alternatively, add to the caption a reference to the additional information available in Table 1.

RESPONSE: The coordinates for all unnamed glaciers (b-f) are included in Table 1.

144-149: It is confusing to use the term "landform" when you mean "glacier". Unless the authors want to refer to different sections of a glacier but with different surface types, however, if that is the case, it does not make sense to say that the insulated part of Tapado Glacier is insensitive to environmental change while its accumulation area is a sensitive "landform".

RESPONSE: We agree and have changed all instances of "landform" to "glacier."

159: "It is likely" seems a euphemism for something that unquestionably will happen.

RESPONSE: We have deleted this sentence.

Table 1: What is the point of comparing this article classification with DGA/IANIGLA classification? Each of these is classifying completely different attributes of the glacier: Glacier sensitivity to environmental change in this article, glacier shape/main characterizing feature for DGA, and glacier debris cover for IANIGLA.

RESPONSE: A major motivation for including these classifications is precisely to show that the classification schemes are different between the two countries and that the scheme used by the DGA is not very helpful for evaluating water resources. We elaborate on these classifications and their usefulness for evaluating water resources in section 4.

214: Which are the distinct hydrological roles? The authors only point to differences in the timescales and the degree to which these glacier types play a role as water reservoirs.

RESPONSE: We have explicitly defined the hydrological role in the first paragraph of the introduction.

"Here we define hydrological role as including contributions to the catchment as well as the impact on storage and drainage of water. For example, glaciers that are more sensitive to changes in climate (e.g. debris-free glaciers) provide a relatively large annual contribution to streamflow now, while rock glaciers are less sensitive and provide a longer-term reservoir (Jones et al., 2018), in some cases even acting as perched aquifers (De Pasquale et al., submitted)."

While strictly speaking if the hydrological role is defined as a particular function within an ecosystem, differences in timescales of water contribution (short-term versus long-term) represent the same hydrological role. However, we would like to continue to use the term "hydrological role" since this term has been used to describe differences in timescales in previously published papers on the subject of rock glaciers and water resources (e.g. Jones et al. 2018; De Pasquale et al., submitted; Schaffer et al., 2019). If the reviewer feels strongly about not using this term please suggest an alternate term. We would be open to using it.

227-229: While that might be more objective, it seems a nightmare from a legal point of view. One can picture a development project affecting a sensitive glacier because a logistic regression happens to assign it to the wrong category.

RESPONSE: We agree and have removed the suggestion to use logistical regression. Instead, we have proposed other quantitative methods that are more appropriate.

256-257: As for line 214, it seems that "role" is not the best word to distinguish between the hydrological effects of different types of glaciers.

RESPONSE: We have explained why we use the term "hydrological role" in the response to the comment on line 214.

**References:**

Ayala, Alvaro, Pellicciotti, Francesca, MacDonell, Shelley, McPhee, James and Burlando, P. (2017) Patterns of glacier ablation across North-Central Chile: Identifying the limits of empirical melt models under sublimation-favorable conditions. Water Resources Research, 53 (7). pp. 5601-5625. ISSN 0043-1397

Ayala, A., Pellicciotti, F., MacDonell, S., McPhee, J., Vivero, S., Campos, C. and Egli, P.: Modelling the hydrological response of debris-free and debris-covered glaciers to present climatic conditions in the semiarid Andes of central Chile, Hydrol. Process., 30(22), 4036–4058, doi:10.1002/hyp.10971, 2016.

Bradley RS, Vuille M, Diaz HF, Vergara W (2006) Threats to water supplies in the tropical Andes. Science 312(5781):1755–1756. https://doi.org/10.1126/science.1128087

Burger, F., Ayala, A., Farias, D., Shaw, T. E., MacDonell, S., Brock, B., ... & Pellicciotti, F.: Interannual variability in glacier contribution to runoff from a high-elevation Andean catchment: understanding the role of debris cover in glacier hydrology. Hydrological Processes, 33(2), 214-229, 2019.

De Pasquale, G., Valois, R., Schaffer, N. and MacDonell, S.: Geophysical signature of two contrasting Andean rock glaciers, Crosph., submitted.

Huss, M. and Hock, R.: Global-scale hydrological response to future glacier mass loss, Nat. Clim. Chang., 8(2), 135–140, 325 doi:10.1038/s41558-017-0049-x, 2018.

Jones, D. B., Harrison, S., Anderson, K. and Betts, R. A.: Mountain rock glaciers contain globally significant water stores, Sci. Rep., 8(2834), doi:10.1038/s41598-018-21244-w, 2018.

MRI Working Group (2015) Elevation-dependent warming in mountain regions of the world. Nat Clim Chang 5:424–430. https://doi.org/10.1038/nclimate2563

Rounce, D. R., Hock, R., McNabb, R. W., Millan, R., Sommer, C., Braun, M. H., Malz, P., Maussion, F., Mouginot, J., Seehaus, T. C. and Shean, D. E.: Distributed Global Debris Thickness Estimates Reveal Debris Significantly Impacts Glacier Mass Balance, Geophys. Res. Lett., 48(8), e2020GL091311, doi:10.1029/2020GL091311, 2021.

Souvignet M, Gaese H, Ribbe L, Kretschmer N, Oyarzún R (2010) Statistical downscaling of precipitation and temperature in northcentral Chile: an assessment of possible climate change impacts in an arid Andean watershed. Hydrol Sci J 55(1):41–57. https://doi.org/10.1080/02626660903526045

Schaffer, N., MacDonell, S., Réveillet, M., Yáñez, E. and Valois, R.: Rock glaciers as a water resource in a

changing climate in the semiarid Chilean Andes, Reg. Environ. Chang., 19, 1263–1279, doi:10.1007/s10113-018-01459-3, 2019.

---

## Author Comment (AC3)

**Reviewer 3**

Schaffer and MacDonell present a new classification of glaciers in terms of their hydrological importance. The new classification is proposed in the context of the ongoing discussion of the Glacier Protection Law (GPL) in Chile and how this classification could be useful in defining the level of protection. The classification is determined in terms of its sensitivity to environmental changes based on the percentage of debris-cover area and surface characteristics. Based on this, the authors mention that this classification could allow different degrees of protection depending on their sensitivity which, is closely related to the hydrological role. The classification includes debris-free glaciers (highly sensitive and greater contribution to streamflow), debris-cover glaciers, including thermokarst features and zones of exposed ice in the surface (semi-sensitive) and rock glaciers, classified as insulated from the environment and hence with a lower hydrological role. Some examples are given for the semiarid environment in Chile and Argentina. The manuscript is well written and could be an important contribution to the discussion of GPL but also to understand the hydrological role of glaciers and its differences.

My main concern, however, is related to the classification and the criteria used and how useful could be to GPL discussion. I understand this as a highly complex topic and probably there could be several classifications criteria depending on the researchers. So please, take my comments on this topic as recommendations. Having said this, I recommend clarifying some points in the manuscript in order to understand the proposed classification.

**General comments:**

*Classification*

My general view is that the classification is simple and do not captures the diverse nature of glaciers in the Andes. In L115-116 it is mentioned that the guidelines need to be evaluated on a case-by-case basis. If this is the case, and I agree with previous reviewers, things seems to be more complicate following this classification. The hydrological role of a glacier, it is probably a concept that most of non-experts understand as a key role of the glaciers. In this sense, an explanation of the differences in time responses it is more adequate to introduce in a context of a GPL discussion in order to fully protect all glaciers. However, I agree that this classification can be useful to water resource management (L66-68).

RESPONSE: We agree that the qualitative approach proposed here is simplistic compared to the heterogeneity and variability that exist among glaciers. We envision the methodology outlined in this paper as an initial classification that could be efficiently completed at a national scale using data already available (e.g. high resolution satellite imagery). In the paper we now suggest that a more sophisticated and quantitative approach that could consider topography, climate, anthropogenic factors such as black carbon be applied as the data, advancements in methodology required, and qualified personnel become available. However, this approach would require much more time, expert professionals and in situ data, so it may be challenging given that there are no trained glacier professionals in the EIA system or local government departments in Chile and there is very limited in situ data available to complete a more sophisticated and quantitative modelling approach at a regional scale. We have modified the discussion paragraph starting on

line 227 to suggest this two-tiered approach (an initial classification as outlined in this paper, followed by a more quantitative and sophisticated approach). We have also modified and expanded upon the quantitative approaches suggested. Finally, we have explicitly identified the limitations of the quantitative approach presented in this paper at the beginning of this paragraph (line 228).

We have added a paragraph at line 49 discussing the large variation in climate, topography, and glacier characteristics that exists from north to south in the Andes and recognize that the dividing line (debris thickness threshold between categories) will vary from north to south. We clarify here and, in the discussion, that the study area chosen is meant to function as an example upon which classification schemes for other regions could be based. We have added a new paragraph starting on line 243 that details how the dividing line might vary from north to south and why with an emphasis on the difference between the semiarid Andes and Patagonia..

A simple modeling approach could be applied such as a temperature-index model that includes solar radiation. However, above 4000 m a.s.l. the performance of temperature-index models is poor within the study area (Ayala et al., 2017). Additionally, this type of model would not be able to incorporate debris thickness and would therefore not provide realistic results for sensitivity. A debris-cover model would need to be used to calculate the thickness, then this would need to be incorporated into a mass balance model capable of accounting for debris-cover. A global debris-cover thickness model only requiring input data that can be obtained remotely (geodetic mass balance and velocity fields) has been developed and these outputs could be used to help differentiate between sensitive and semi-sensitive landforms (Rounce et al., 2021). The outputs from an earlier version of this model compare well to measurements of debris thickness on Pirámide Glacier (Ayala et al., 2016), but comparison with other glaciers in the semiarid Andes is necessary to evaluate the accuracy since the model was calibrated on a debris-covered glacier in Nepal. At present, methods for modelling thick debris cover (e.g. > 2 m) have not been validated and are therefore not a reliable tool to differentiate between semi-sensitive and insulated landforms.

While strictly speaking if the hydrological role is defined as a particular function within an ecosystem, differences in timescales of water contribution (short-term versus long-term) represent the same hydrological role. However, we would like to continue to use the term "hydrological role" since this term has been used to describe differences in timescales in previously published papers on the subject of rock glaciers and water resources (e.g. Jones et al. 2018; De Pasquale et al., submitted; Schaffer et al., 2019). If the reviewer feels strongly about not using this term please suggest an alternate term. We would be open to using it.

We have modified the first paragraph in the introduction to highlight the usefulness of the classification scheme proposed with respect to legislation in general, including the GPL. We have also added a few sentences to the discussion starting on line 257 to highlight how incorporating the classification scheme in combination with water-scarcity could improve the GPL by making it possible to match the level of protection to the water resource need resulting in protection that would be region-specific, meet the needs of society without over- or under-protecting, and could evolve through time as the climate and water availability changes. We agree that introducing the proposed classification would likely complicate the proposed GPL and

make it more difficult to pass this law. However, the currently proposed GPL is limited in its ability to effectively protect glaciers as a single classification for all glaciers makes it rigid in both space and time. We have also added a couple sentences to the first paragraph of the introduction to highlight the usefulness of the proposed classification for glacier management (EIA).

*Concepts*

The concept of hydrological role is not well defined in the manuscript. Following the explanation in L144-157, it seems that it is related mainly to the contribution of each type of glacier to streamflow at an annual scale. I suggest a clearer definition of what the authors mean by "hydrological role" including temporal and spatial scales and also the potential contribution to groundwater. The authors mentioned (L30-33 and L155-157) that insulated glaciers (rock glaciers) storage and delay the runoff. This is an important point as the hydrological role and importance of these glaciers have a different time scale in comparison to debris-free glaciers and must be included to define the hydrological role.

RESPONSE: We have explicitly defined the hydrological role in the first paragraph of the introduction.

"Here we define hydrological role as including contributions to the catchment as well as the impact on storage and drainage of water. For example, glaciers that are more sensitive to changes in climate (e.g. debris-free glaciers) provide a relatively large annual contribution to streamflow now, while rock glaciers are less sensitive and provide a longer-term reservoir (Jones et al., 2018), in some cases even acting as perched aquifers (De Pasquale et al., submitted)."

"Sensitivity to environmental changes" I understand that probably the use of "environmental changes" is used to include the atmospheric drivers of the melt as well as the feedback (positive or negative) that the debris-cover and glaciers surface characteristics exert on melt rates. However, the concept of "environmental changes" is wide and includes several other factors. I suggest clarifying what exactly means "sensitivity to environmental changes". Maybe, constrain this concept will allow a clearer link between the classification and the hydrological role. I think in a concept like "sensitivity to melt drivers" or probably something better.

RESPONSE:In the introduction we now define sensitivity to environmental changes as including temperature, precipitation, and black carbon.

*Clarification on the level of protection*

In order to avoid confusion, I suggest including, explicit, the order of the level of protection for each type of the classification i.e. the type that needs more protection according to your classification.

RESPONSE: We are proposing a type-dependant level of protection. We have added a paragraph to the discussion starting on line 260 that addresses the level of protection. We think that the specific decisions with regards to the level of protection for each region and assigned to each

glacier category proposed are public policy decisions that require balancing many factors such as water resources and the economy and are beyond the capacity of authors of this paper to decide. However, we do provide some general recommendations.

**Specific comments:**

L20 "Sendado" to "Senado" (also in the reference list).

RESPONSE: This change has been made.

L274-278: Although is not the focus of the paper, I think that the other values of glaciers mentioned here must be included in the Introduction. This manuscript is concentrated on the meltwater contribution to runoff, which of course is important, but as mentioned, glaciers also play other key roles.

RESPONSE: The majority of the additional roles are already mentioned in the introduction when defining what glacier protection laws aim to preserve (glaciers as strategic water reserves, for their role in sustaining biodiversity, in sustainable tourism, and their scientific importance). The other roles mentioned on lines 274-278 are very important, and we have incorporated these into the introduction.

**References:**

Ayala, Alvaro, Pellicciotti, Francesca, MacDonell, Shelley, McPhee, James and Burlando, P. (2017) Patterns of glacier ablation across North-Central Chile: Identifying the limits of empirical melt models under sublimation-favorable conditions. Water Resources Research, 53 (7). pp. 5601-5625. ISSN 0043-1397

Ayala, A., Pellicciotti, F., MacDonell, S., McPhee, J., Vivero, S., Campos, C. and Egli, P.: Modelling the hydrological response of debris-free and debris-covered glaciers to present climatic conditions in the semiarid Andes of central Chile, Hydrol. Process., 30(22), 4036–4058, doi:10.1002/hyp.10971, 2016.

Bradley RS, Vuille M, Diaz HF, Vergara W (2006) Threats to water supplies in the tropical Andes. Science 312(5781):1755–1756. https://doi.org/10.1126/science.1128087

Burger, F., Ayala, A., Farias, D., Shaw, T. E., MacDonell, S., Brock, B., ... & Pellicciotti, F.: Interannual variability in glacier contribution to runoff from a high-elevation Andean catchment: understanding the role of debris cover in glacier hydrology. Hydrological Processes, 33(2), 214-229, 2019.

De Pasquale, G., Valois, R., Schaffer, N. and MacDonell, S.: Geophysical signature of two contrasting Andean rock glaciers, Crosph., submitted.

Huss, M. and Hock, R.: Global-scale hydrological response to future glacier mass loss, Nat. Clim. Chang., 8(2), 135–140, 325 doi:10.1038/s41558-017-0049-x, 2018.

Jones, D. B., Harrison, S., Anderson, K. and Betts, R. A.: Mountain rock glaciers contain globally significant water stores, Sci. Rep., 8(2834), doi:10.1038/s41598-018-21244-w, 2018.

MRI Working Group (2015) Elevation-dependent warming in mountain regions of the world. Nat Clim Chang 5:424–430. https://doi.org/10.1038/nclimate2563

Rounce, D. R., Hock, R., McNabb, R. W., Millan, R., Sommer, C., Braun, M. H., Malz, P., Maussion, F., Mouginot, J., Seehaus, T. C. and Shean, D. E.: Distributed Global Debris Thickness Estimates Reveal Debris Significantly Impacts Glacier Mass Balance, Geophys. Res. Lett., 48(8), e2020GL091311, doi:10.1029/2020GL091311, 2021.

Souvignet M, Gaese H, Ribbe L, Kretschmer N, Oyarzún R (2010) Statistical downscaling of precipitation and temperature in northcentral Chile: an assessment of possible climate change impacts in an arid Andean watershed. Hydrol Sci J 55(1):41–57. https://doi.org/10.1080/02626660903526045

Schaffer, N., MacDonell, S., Réveillet, M., Yáñez, E. and Valois, R.: Rock glaciers as a water resource in a

changing climate in the semiarid Chilean Andes, Reg. Environ. Chang., 19, 1263–1279, doi:10.1007/s10113-018-01459-3, 2019.

---

## Author Response (AR2)

Harry Zekollari,

Thank you for all the good publication suggestions, thoughtful questions, and thorough review of the text. Incorporating the suggested changes has made the paper much stronger.

- Nicole Schaffer and Shelley MacDonell

**Editor**

**General comments**

First, I would like to apologize for the delay in finalizing the editing of this manuscript. I would like to thank both authors for having addressed the comments by the reviewers and for having updated the manuscript accordingly. The brief communication is now in a good shape to almost be accepted for publication. I have formulated a series of, mostly minor, comments that I invite the authors to address before we can advance to a full acceptance of this manuscript.

**Specific comments**

- l.8: "However, these laws are limited…" i.e. suggest adding 'laws' to clarify.

RESPONSE: We have modified the text as requested.

- l.9: "…monitor water resources…", i.e. suggest removing 'these'.

RESPONSE: We have modified the text as requested.

- last sentence of abstract: found this to be a bit surprising as a kind of statement and a bit disconnected with respect to the rest of the abstract. Maybe connect with: "Finally, we also review both national…"?

RESPONSE: We have modified the text as requested.

- l. 19: "specifically designed for glaciated regions", i.e. suggest adding 'designed'.

RESPONSE: We have modified the text as requested.

- l. 27: Andean glaciers and their importance as national heritage. Suggest also adding a reference to Bosson et al. (2019) here, which also includes glaciers over your study area

RESPONSE: The reference has been added.

- l.34-35: "Here, we define the hydrological role of the glacier as including contributions…": suggest making this more specific.

RESPONSE: We have removed this sentence in response to the comment for l.36.

- l.36: "glaciers that are more sensitive to changes in climate (e.g. debris-free glaciers)": this is a central statement in your story. But is this really the case? Are debris-free glaciers changing more than debris-covered glaciers? Suggest adding references to back this up. For instance, the recent changes for all glaciers in South America are known: is it clear that debris covered glaciers change less than debris free ones? For this, refer to two important studies by Braun et al. (2019) and Dussaillant et al. (2019), which are complete and allow you to quantify this statement. Or rely on the recent global product by Hugonnet et al. (2021).

RESPONSE: Thank you for your questions regarding debris-free versus debris-covered glaciers. We have reviewed the literature on this topic and find that elevation change data does support the theory of debris-free glaciers changing more than debris-covered glaciers for the La Laguna catchment (Robson et al. 2022) and this agrees with a global study by Rounce et al. (2019) who conclude that the net effect of accounting for debris in all regions is a reduction in sub-debris glacier melt, by 37% on average. However, this does not hold true everywhere in the semiarid Andes nor in the world. For example, Ayala et al. (2016) report similar mass losses for Pirámide glacier (classified as intermediate) and two nearby debris-free glaciers, mainly because Pirámide is at a lower elevation. Similar mass losses have also been observed in the Himalaya (Gardelle et al. 2013; Kääb et al., 2012). We suggest a conservative approach when assigning a level of sensitivity for protection to intermediate glaciers (previously called semi-sensitive glaciers in the text) by initially assuming they will have the same mass balance rate as sensitive glaciers, with the option to downgrade this if there is data available to justify the change. For a more complete discussion please see the modifications made to the text starting at line 149.

We have added a table with some examples of intermediate glaciers that were compared to sensitive and/or insulated glaciers using elevation change data and assigned a (revised) category for protection using the method described after line 142 (third paragraph; see Table 2). For these examples we use the data sets from Ayala et al., (2016), Robson et al. (2022) and Braun et al. (2019). The study area is also covered by Hugonnet et al. (2021) and Dussaillant et al. (2019), but these data sets are only available for download as tiles with a resolution of 1 degree, which is too coarse for this comparison.

Through this literature review we have understood that glacier type does not necessarily correspond to the hydrological role (e.g. sensitive and intermediate glaciers may have the same mass balance rate). We have therefore removed the association between sensitivity and the hydrological role in the paper. The sentence on line 35 and the rest of this paragraph has been modified to reflect that and better define sensitivity. We have also made changes throughout the text.

- l. 40: "…reflects their sensitivity, which is closely related to their hydrological role": again, need statement to back this up. Also refer to previous comment for this.

We have modified this sentence to only refer to the sensitivity (please see our comment for l.36)

- l.44 and throughout the manuscript (e.g. l.260, l.288, l.292,…): "black carbon" is mentioned many times. But does this really play a big role for glaciers here? Is this from local sources/roads? And how can this influence the mass balance of debris-covered and rock glaciers? Would suggest mentioning this less explicitly/often, as expect it is of (very) minor importance for glacier mass balance. Or would need references to back this up.

RESPONSE:  We agree that black carbon is over-emphasized in the text. The impact on mass balance in the study region is largely unknown as there are no peer-reviewed published papers on this specific topic in the area. We have added a recently published paper that came out on tracing particulate matter sources for Tapado Glacier and removed most instances of black carbon mentioned in the text. Black carbon is only mentioned once in the introduction and in the discussion.

According to Rowe et al. (2019) the main source of black carbon north of Santiago is emissions for diesel engines that power the mining industry and major astronomical observatories. Near Santiago, sources include transportation (e.g. diesel), industrial pollution, and residential heating. The regional average vertically-integrated loading of black carbon is much lower in the north compared to further south, but albedo reductions measured in snow due to light absorbing impurities is higher in the north (Rowe et al., 2019). We do not expect that black carbon has an important influence on debris-covered glaciers that are fully covered or rock glaciers.

- l.47-49: "Interpretations range from…": not entirely clear what refers to what in this sentence. "a glacier that has a very thin debris cover" refers to debris covered glacier and "thick enough debris cover to insulate the ice below" to rock glacier?

RESPONSE: We have modified this sentence to clarify as follows:

"In some instances glaciers that have a very thin debris cover and some ice exposed are considered rock glaciers (e.g. Chilean national inventory), while in other cases a thick enough debris cover to insulate the ice below is required (~> 3 m; Janke et al., 2015)."

- l.50-51: sensitive and insensitive to environmental changes: if this is the case, would need to see this in observed glacier changes (which should correlate to degree of debris cover and thickness), is this the case from e.g. Braun et al. (2019) and Dussaillant et al. (2019) for glaciers in the Andes?

RESPONSE:  We have modified this sentence to refer specifically to debris cover and not to sensitivity in general and it now reads: "The difference between these interpretations is an important consideration since the former option potentially encompasses glaciers that have a debris cover thin enough to allow sufficient heat transfer to melt the ice surface below (e.g. < 0.2 m; Nicholson and Benn, 2006),  while the latter option only includes glaciers that have a thick enough debris cover to, in theory, insulate them from heat at the surface (Bonnaventure and Lamoureux, 2013; Janke et al., 2015)."

We were able to compare the glacier Las Tetas to both Tapado and an insulated glacier nearby using the output from Robson et al. (2022) and this shows Tapado (debris-free part) with the

greatest mass loss rate, followed by Las Tetas (intermediate glacier), followed by the insulated glacier. This is included in the examples in Table 2 and in the .kmz file. We were also able to do a comparison with one glacier in the Braun et al. (2019) data set and found the same pattern. However, this dataset generally excludes rock glaciers so only a small portion of the insulated glacier could be compared.

- l.57-60: found sentence hard to understand. Suggest splitting in two sentences and being more specific [adding "these inventories"]: "… with the proposed groups. Based on this, suggestions are provided to modify these inventories to facilitate…"

RESPONSE: We have modified the text as requested.

- l. 64: "which identifies four distinct zones": not entirely clear which these zones are. Below you mention three zones (which partly overlap by the way). I may be missing the point / misunderstanding, but suggest making this more consistent: e.g. by mentioning the 4 zones, and then mentioning on which you decide to focus.

RESPONSE: We have modified the text to clarify this by mentioning all four zones then specifying the region we are focusing on.

- l.71: water availability. Suggest to possibly make link here with the Water Tower Index (or more specifically the 'Supply Index') by Immerzeel et al. (2020)

RESPONSE: We have added a sentence to describe the Water Tower Supply Index on line 67.

- l.110: definition of the threshold at 0.3 m. Is this based on the single study mentioned in the line before? Would be good to clarify.

RESPONSE: The 0.3 m threshold is primarily based on the Ferrando (2012) and Ayala et al. (2016). Direct measurements in Ferrando (2012) show that persistent surface melt occurs at 0.3 m debris thickness which indicates the threshold should be > 0.3 m. The modelled debris cover thickness and mass balance in Ayala et al. (2016) for Piramide roughly agree with this threshold. Mass balance becomes more negative as elevation decreases as would be expected, until ~3800 m.a.s.l, below which debris cover thickens, and the mass balance suddenly becomes less negative and remains constant down-glacier (~-1 m w.e. a-1). The debris thickness at 3800 m a.s.l. is heterogeneous with a range of approximately 0.1-0.5 m thick (modelled debris thickness).

Your review comment prompted me to contact the author A. Ayala to ask if he could provide additional data that would help to identify an appropriate threshold. He sent me some plots of modelled debris thickness versus mass balance plus an interpretation of these. These plots of modelled debris thickness versus mass balance show that on Pirámide ablation is reduced by 80% when debris thickness is 30 cm and 90% when it is 60 cm (A. Ayala, personal communication, March 7 2022). He also mentioned that modelled debris thicknesses > 0.2 m in this study under-estimate compared to in situ measurements and are prone to error so these results should be interpreted with caution. This agrees with Rounce et al. (2021) who provide globally distributed debris thicknesses and sub-debris melt outputs and conclude that thin debris

cover (typically 0.03 m – 0.05 m) enhances sub-debris melt while thick debris cover can result in a >90% reduction in sub-debris melt. Based on these results we feel it is reasonable to be more specific with the threshold and place it at ~0.5m debris thickness (as opposed to > ~0.3m). We have updated the text to reflect this change and have added additional information from the study by Ayala et al. (2016) and personal communication with A. Ayala.

A threshold of > 0.3 m and ~0.5m roughly agrees with modelled debris cover thickness and mass balance results on Bello and Yeso Glaciers that are adjacent to Piramide as well (Ayala et al. 2016). On Bello the vast majority of the debris cover is < 0.2 m (modelled thickness). For this glacier mass balance has a linear relationship with altitude and the debris cover near the terminus has a minimal effect on the mass balance pattern. On Yeso glacier there is a thick debris patch at the terminus (0.2-0.6 m) and this is a associated with a very obvious decrease in mass balance at the terminus.

- l.113: "about 95% of the surface", or more, right?

RESPONSE: Yes, we have modified the sentence adding "or more."

- l.114-115: using the surface cover as a proxy for debris cover thickness seems to be relatively rough / qualitative.. Especially given that this seems to rely on a single study. Possibility to make this statement sounder / adding additional studies to support this? Here, without knowing a lot about debris, would think of e.g. work by Scherler et al. (2018), Herreid and Pellicciotti (2020) and Rounce et al. (2021). Especially the latter seems important and would be very relevant in your story in general, as it estimates the debris thickness for all glaciers. I see that you briefly refer to this study later on, but it would be good to have this more prominently featured. From my understanding, despite some of the limitations (you mention the debris thickness is derived from relationship from glacier in High-Mountain Asia), it would be a great tool that could maybe directly / or in complement with what you suggest here, be used to categorize glaciers in terms of debris presence/thickness (and related sensitivity to climate change, which you mention)

RESPONSE: In response to this comment, we have added the following sentence:

"Global products of glacier debris cover could be used to quantify the percentage of debris cover to remove subjectivity (e.g. Herreid and Pellicciotti, 2020; Scherler et al., 2018), however outputs have not been validated for the Andes and coverage is limited to glaciers included in the RGI. We proposed that this initial classification could be refined or used in combination with modelled debris thicknesses (e.g. Rounce et al., 2021) but not replaced by these model outputs since these have not been validated for the Andes and coverage is limited (see Sect. 5)."

We have also added the following sentence on L141:

"Differentiation between intermediate and insulated glaciers could be made more robust by combining the qualitative classification with modelled debris thicknesses, but not be replaced given that methods for modelling thick debris cover (e.g. > 2 m) have not been validated (see Sect. 5)."

- l.132: permafrost in the glacier. Again, probably related to my relatively limited knowledge about the subject: but can there be permafrost in a glacier? Or is this specifically for rock glaciers (which you seem to target under the next category, nr.3)

RESPONSE: We have removed the word permafrost.

- l.140-169: quite long and found this to be disturbing the flow of your manuscript a bit. Could you consider slightly shortening this?

RESPONSE: We have made this section more concise.

- l.193: "These glaciers are more responsive to climatic changes". See also previous remark on this (l.36): this is quite central in your story, but is this also clear from large datasets that cover your glaciers of interest (Braun et al., 2019; Dussaillant et al., 2019; Hugonnet et al., 2021)? Moreover, also the study by Rounce et al. (2021) could help answer this, as the title of that study suggests: "Distributed global debris thickness estimates reveal debris significantly impacts glacier mass balance"

RESPONSE: Please see our answer to the review comment for l.36. We have also modified this paragraph significantly.

- l.201: "such as contructions of roads": could you explain how this affects the glaciers?

RESPONSE: The construction of a road on top of a rock glacier could require the removal or disturbance of ice. Indirect impacts might include the deposition of dust from road construction which can impact glacier mass balance (Rowe et.al. 2019) or the use of heavy machinery (e.g. vibration compactor) which may destabilize the slope and create heat. All of these impacts could lead to permafrost degradation.

- l.209: what are "cryospheric glaciers"?

RESPONSE: We have removed "cryospheric" as this was an error.

- l.224-225: the sentence is not entirely clear. Maybe split in two sentences? e.g. "…(RGI). Insulated glaciers are excluded,…"

RESPONSE: We have divided this in to two sentences.

- l.235: "is completed but not yet publicly available": just checking, is this still the case?

RESPONSE: Yes, this is still the case. The inventory is not yet publicly available.

- l.256-257: "…during a thorough review of the Argentinian inventory". Can this be quantified, is the data possibly available somewhere to show this? Maybe possible to add as supplement to the publication?

RESPONSE: We have included a .kmz file as supplementary material which includes a layer outlining the geographical area reviewed in both the Chilean and Argentinian inventories (a rectangle) as well as all of the examples listed in Table 1. The Chilean and Argentinian inventories are publicly available and the links to access them are provided in the text (we have added the web page where the Chilean inventory can be downloaded). We have also added a reference to the supplementary material at the end of this sentence (supplementary material S1).

- l.270: "not sensitive to environmental changes": ok, way less sensitive than a debris-covered glacier or a debris-free glacier, but in the end also sensitive to environmental changes (although much slower reaction), right?

RESPONSE:  Correct. We have modified the sentence to say "…not very sensitive …"

- l.277: automatic detection methods: suggest mentioning some examples for this here (e.g. Khan et al., 2020; Lu et al., 2021)

RESPONSE: The reference Lu et al. (2021) was added on line 287.

- l.282-283: you suggest this is not a reliable tool. I agree that there are indeed quite some considerable uncertainties, but not sure if it is better to work with relatively qualitative relationship between few measurements debris thickness measurements and link with debris cover..

RESPONSE: The errors with estimating thick debris cover are large (e.g. ~0.2 m according to Ayala et al., 2016), however the error from the qualitative relationship could definitely be large as well. We have modified this sentence in light of this comment.

"At present, methods for modelling thick debris cover (e.g. > 2 m) have not been validated so their effectiveness at differentiating between intermediate and insulated glaciers is unknown."

Given the uncertainty in both methods, perhaps a convergent approach would work where both methods are used. If they agree a high degree of confidence is assigned to the classification. If they disagree, a low degree of confidence is assigned to the classification. We have modified the sentence starting on L275 to reflect this

"We therefore propose that this be used as an initial classification which is later refined or used in combination with a more sophisticated…"

- l.298-314, l.317-324, l. 348-356: quite long and not very specific. In some cases, also quite repetitive. It would be good to shorten these passages.

RESPONSE: We have modified these sections to remove repetitive and unnecessary text.

- l.317-318: not sure if I understood this correctly. Are there many categories in Janke et al.? From this, various categories were removed to have 2 categories, after which 'additionally' debris-free glaciers were added to have 3 categories in the end?

RESPONSE: Yes, that is correct. We have modified these sentences to clarify as follows:

"These categories are aligned with Janke et al. (2015) who propose six categories for debris-covered and rock glaciers. The categories in this paper additionally include debris-free glaciers and the number of categories has been reduced to three."

- l.365-368: as there is a need for time and expertise to apply this, would it not make sense to work with automated products, which are in some cases directly available? (e.g. Rounce et al., 2021)

RESPONSE: The distinction between sensitive and intermediate glaciers could be completed with an automatically generated product (e.g. Rounce et al., 2021), but these outputs should be compared to measured debris thicknesses on glaciers in the semiarid Andes to evaluate their accuracy since the model was calibrated on a debris-covered glacier in Nepal. Furthermore, this study uses the Randolph Glacier Inventory. This inventory does not include insulated glaciers and for the vast majority of hybrid glaciers (e.g. sensitive and semi-sensitive) only a small portion of the glacier is included if ice is exposed otherwise these glacier types are excluded as well so the automatic product by Rounce et al. (2021) would be missing a large number of glaciers/parts of glaciers. Despite these limitations we agree that it would be beneficial to incorporate automated products and we have therefore modified the text on L114 as follows:

"…Therefore, having > 95 % of the surface or more covered by debris could be used as a criterion to approximately identify this threshold using satellite imagery. Global products of glacier debris cover could be used to quantify the percentage of debris cover to remove subjectivity (e.g. Herreid and Pellicciotti, 2020; Scherler et al., 2018), however outputs have not been validated for the Andes and coverage is limited to glaciers included in the RGI. We proposed that this initial classification could be refined or used in combination with modelled debris thicknesses (e.g. Rounce et al., 2021) but not replaced by these model outputs since validation in the Andes is needed and coverage is limited (see Sect. 5)."

At present, methods for modelling thick debris cover (e.g. > 2 m) have not been validated so their effectiveness at differentiating between intermediate and insulated glaciers is unknown. However, we do agree that incorporating modelled debris cover would be beneficial. We have added the following sentence at L 141:

"Differentiation between intermediate and insulated glaciers could be improved by using both the qualitative classification proposed and modelled debris thicknesses, although these model outputs have large uncertainties (see Sect. 5)."

**References**

All references mentioned above are now included in the reference section of the manuscript.